# ContPhy: Continuum Physical Concept Learning and Reasoning from Video

## Abstract

We introduce the Continuum Physical Dataset (ContPhy), a novel benchmark for evaluating machine models in physical reasoning across diverse scenarios for the continuum. The ContPhy is specifically designed to be complementary with existing physical reasoning benchmarks by encompassing diverse physical property inferences for the physical parameters of the continuum such as mass and density across dynamic situations and predicting corresponding dynamics. This comprehensive dataset enables the development and assessment of AI models with human-like visual reasoning abilities in understanding visual attributes, physical properties, and dynamics of both rigid objects and soft objects[1] while devising goal-oriented solutions. We evaluated a range of AI models and found that they still struggle to achieve satisfactory performance, which shows that current AI models still lack physical commonsense for the continuum, especially soft-bodies, and illustrates the value of the proposed dataset. We hope the ContPhy fosters advancements in AI perception and reasoning in diverse physical environments, bridging the gap between human and machine intelligence in the physical world [2].

## 1 Introduction

Humans are capable of comprehending the physical properties of various substances, including rigid objects and soft objects, understanding their dynamic interactions in complex environments, and predicting their corresponding dynamic changes. In fact, this innate ability to understand and reason about the physical world plays a crucial role in shaping our understanding of nature and the development of scientific knowledge.

As depicted in Fig. 1, objects like solids and liquids in nature often exhibit different properties, and these objects of different properties couple together to build our complex physical world. As humans, we are able to distinguish objects' physical properties by observing their interactions. We are able to know that the clear liquid in Fig. 1 (a) at the bottom has a higher density than the yellow liquid on the top; we know that the dynamic pulley in Fig. 1 (c) could help us to pull the cargo up more easily. These innate human skills raise an intriguing question: can current AI models have the physical common sense to infer physical properties of the continuum and predict corresponding dynamics?

Recently, a series of benchmarks (Riochet et al., 2018; Rajani et al., 2020; Bear et al.), have been developed to study machine models' effectiveness at physical reasoning. However, there have been limitations that make them non-ideal benchmarks for the development and assessment of whether machine models have human-like visual reasoning abilities in understanding objects' physical properties and dynamics. Firstly, most of the benchmarks mainly deal with simple visual primitives like spheres, cubes and collision events of rigid objects only. It remains doubtful whether the conclusions based on these simple scenes will still hold in more comprehensive visual scenarios with the coupling of soft objects and their interaction with rigid objects. There have also been benchmarks like Physion (Bear et al.) that were developed to evaluate machine models' physical reasoning abilities. However, objects in Physion are of the same physical parameters without any variance (*e.g.* solids with the same mass and water with the same density). Moreover, the Physion dataset

---

[1]Continuum encompasses various bodies like liquids, soft materials (*e.g.*, soft balls, cloth, and ropes), rigid bodies (*e.g.*, cubes, pillar, plate and spheres), and articulated bodies (*e.g.*, pulleys)

[2]Project page: `https://physical-reasoning-project.github.io`

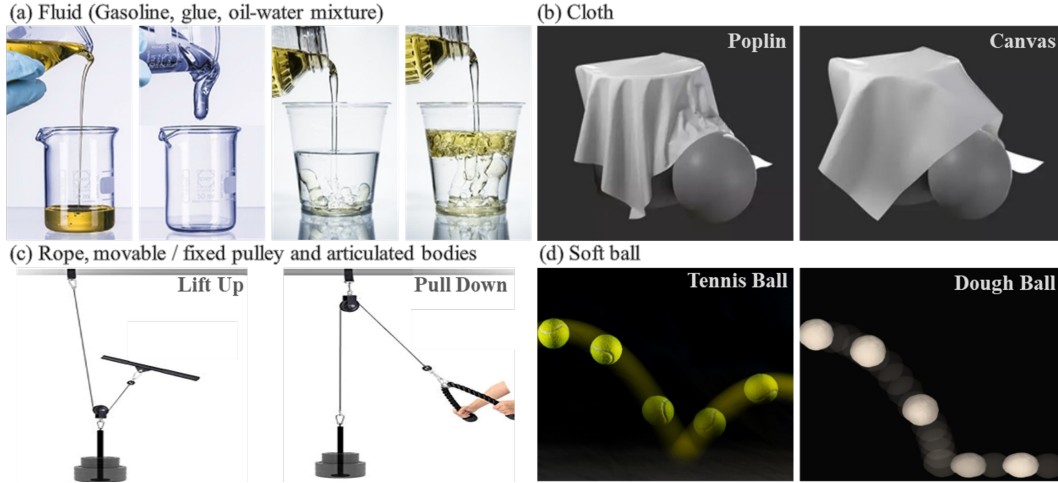

Figure 1: The motivation is derived from a range of everyday soft materials and their interaction with rigid objects, whose physical behaviors or functions vary by their diverse physical properties. a) Gasoline flows more fluently than glue due to lower viscosity, while oil with lower density tends to float above water. b) Poplin and canvas exhibit surface wrinkles with varying granularity due to their distinct bending compliance. c) The lifting approach requires less force due to the re-distributed tensile forces facilitated by the movable pulley. d) Trajectories of tennis ball and dough ball demonstrate their differing elasticity and plasticity.

only requires models to predict whether two objects will come into contact after the observed video ends. It has not incorporated natural language to answer other challenging questions like predicting dynamics in counterfactual scenes and selecting actions to achieve a goal.

To this end, we aim to build a Continuum Physical Dataset (ContPhy) to thoroughly evaluate and diagnose machine models' physical reasoning performance in comprehensive physical environments. The design of ContPhy aims to achieve two goals: 1) Cover diverse physical scenarios and 2) support comprehensive natural language tasks. To achieve the first goal to build diverse physical scenes, we adopt the physical engine Unity (Haas, 2014) to simulate diverse videos with dense supervision signals. As shown in Fig. 2, the simulated physical scenes include scenes with the coupling of different liquids, scenes with deformable cloths and rigid objects, and scenes with pulley systems. Another goal of the built dataset is to propose diverse physical reasoning tasks in the form of video question answering. We achieve this goal with a carefully-designed question generator. The question engine takes the dense simulated video annotation as input and generates different questions based on pre-defined textual templates. Sample questions can be found at Fig. 2. It asks challenging questions such as "*If the red stick were removed, would most orange fluid flow into the cyan container?*" and "*Is the mass of the sphere greater than half that of the red cube?*", which requires the model to have a deep understanding of physical scenes and reason about their dynamics.

We also evaluate a series of existing machine models (Le et al., 2020; Hudson & Manning, 2018) on the new proposed ContPhy benchmark. We found that the performance of these models is far from satisfactory, demonstrating the proposed ContPhy benchmark's value and indicating the necessity of more advanced machine models with better physical common sense.

To summarize, the contribution of the paper lies in three aspects. First, we propose a new comprehensive physical reasoning benchmark that includes the coupling of different diverse physical properties like mass, density, elasticity, and deformability, and the interaction between soft objects and rigid objects. Second, we build a carefully-designed question generator that is able to synthesize different kinds of challenging physical reasoning question-answer pairs and provides multi-step rationales that lead to the answers. Finally, we extensively evaluate the proposed benchmark with multiple machine models to study the characteristics and show insights into physical reasoning model development.

## 2  RELATED WORK

**Physical Reasoning.** Our work is closely related to Physical Reasoning benchmarks (Riochet et al., 2018; Rajani et al., 2020; Girdhar & Ramanan, 2020; Baradel et al., 2020; Bear et al.; Chen

| Dataset | Question Answering | Rationales | Diverse Scenarios | Goal-driven Questions | Interaction of soft objects | Counterfactual Property Dynamics |
|---|---|---|---|---|---|---|
| IntPhys (Riochet et al., 2018) | × | × | × | × | × | × |
| ESPRIT (Rajani et al., 2020) | × | × | × | × | × | × |
| Cater (Girdhar & Ramanan, 2020) | × | × | × | × | × | × |
| CoPhy(Baradel et al., 2020) | × | × | × | × | × | √ |
| CRAFT (Ates et al., 2020) | √ | √ | × | × | × | × |
| CLEVRER (Yi et al., 2020) | √ | √ | × | × | × | × |
| Physion (Bear et al.) | × | × | √ | × | √ | × |
| ComPhy Chen et al. (2022) | √ | √ | × | × | √ | √ |
| CRIPP-VQA Patel et al. (2022) | √ | √ | × | √ | × | × |
| ContPhy (Ours) | √ | √ | √ | √ | √ | √ |

Table 1: Comparison between ContPhy and other physical reasoning benchmarks. ContPhy is a dataset that covers a wide variety of tasks including reasoning about the continuum's physical properties, counterfactual dynamics, and goal planning in diverse physical scenarios.

et al., 2022; Li et al., 2022b). We summarize the key features of these various benchmarks and compare against our benchmark in table 1. Early benchmarks (Riochet et al., 2018; Rajani et al., 2020) simulate physical scenes with visual primitives and test models' physical intuition. Later, CLEVER (Yi et al., 2020), ComPhy (Chen et al., 2022), and CRIPP-VQA (Patel et al., 2022) extend the simple visual primitives with natural language and asked questions about rigid bodies' collisions. Recently, Physion (Bear et al.) provides more complex visual scenes and requires models to predict whether two objects will come into contact in future frames. As summarized in table 1, the proposed ContPhy is the only benchmark that contains soft objects with different physical parameters and asks diverse language-based questions about dynamics in counterfactual and goal-planning scenarios.

Recently, there is a paper (Li et al., 2022c) raises concerns that dynamics-based models may struggle to make accurate predictions. While several methods (Wu et al., 2022; Ding et al., 2020; 2021; Lu et al., 2023) have successfully tackled previous benchmarks like Physion (Bear et al.) and CLEVRER (Yi et al., 2020). These papers have successfully validated the performance of state-of-the-art AI models. Motivated by their significant contribution, we aim to extend this success further by evaluating dynamics and soft-body objects, and interaction between rigid and soft bodies.

**Visual Question Answering.** Our paper is also related to Visual Question Answering (VQA) (Zadeh et al., 2019; Lei et al., 2019; Wu et al., 2021), which mainly requires machine models to answer questions about a given image or video's content like visual attributes, actions, activity, and social events. VQA was first developed for single images, which mainly asks for objects' categories and visual attributes like color and shapes (Andreas et al., 2016; Hudson & Manning, 2019). Subsequently, it was extended to the video domain. However, these datasets still typically assess abilities in visual perception, recognizing objects, shapes, and colors, and understanding human-centric actions. In this paper, we aim to build a benchmark that could evaluate machine models' comprehensive physical reasoning abilities in scenarios with the continuum, including rigid objects, soft objects and liquids.

**Physical Benchmarks for Soft Bodies.** In recent years, there has been a growing interest in the properties and dynamics of soft-bodied objects within the research community (Xiang et al., 2020; Gan et al., 2020; Macklin et al., 2014; Xian et al., 2023; Haas, 2014). Much of this research has concentrated on creating simulations of deformable objects and fluids using physical engines, thus advancing the field of robotic manipulation and cognitive experimentation. Leveraging these robust tools, we can simulate deformable objects and fluids with varying physical parameters, enabling collaboration with natural language for the purpose of physical commonsense learning. This allows us to investigate the extent to which current AI models comprehend such physical phenomena.

## 3 DATASET

The proposed ContPhy dataset is designed to evaluate machine models' reasoning abilities on comprehensive physical scenes with different objects like rigids, liquids, ropes, and cloths and massive physical properties associated with them. In this section, we describe how we built the dataset. For the various scenarios we propose, a unified data generation pipeline is summarized into 2 key stages, physics simulation and VQA generation. In section 3.1, we introduce how we leverage the simulation engine to build diverse scenes. In section 3.2, we describe how we develop a question

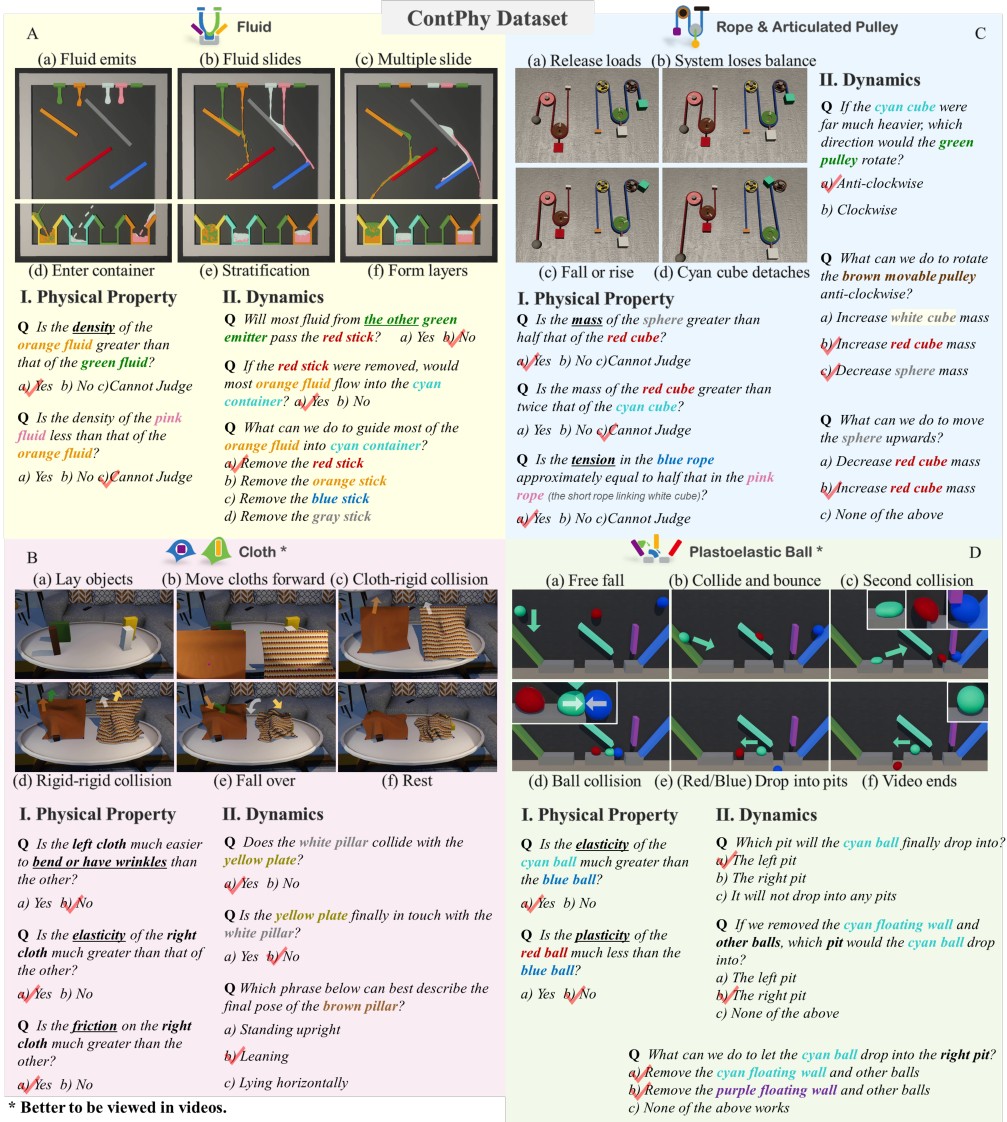

Figure 2: The figure presents samples from the four puzzle blocks of our Continuum Physical Dataset(ContPhy). ContPhy offers rendered outputs from the simulation of randomly sampled scenarios, accompanied by their respective question-answer pairs. These pairs span from understanding soft-body physical properties, concepts and interactions with rigid objects through comparative analysis, to temporal and spatial dynamic predictions, counterfactual considerations, and goal-oriented problem-solving. It aims to provide a comprehensive resource for AI models to interpret the physical world of various deformable bodies.

generator to synthesize diverse questions to test reasoning abilities in different aspects. In section 3.3, we provide statistics about the dataset and how to perform post-processing to reduce data bias.

## 3.1 PHYSICAL DYNAMIC SIMULATION

We used the Unity engine (Haas, 2014) to simulate and render our physical scenes due to its effectiveness and efficiency in simulating physical effects. We design four physical scenes to study different physical phenomena across different object materials with varying physical properties, masses, friction, elasticity, density, deformability, and stretchiness.

**Diverse Object Material.** In the real world, rigid bodies, soft bodies and fluids exhibit different physical properties. Depending on the physical environment, these objects also have different

behaviors. Using the Unity engine, we are able to combine various physical properties compositionally to generate scenes that require a deeper understanding of the physical scenes. For example, as in Fig 1 (c), with the coupling of deformable ropes and pulley systems, we can lift up the cube on the right with less pulling force on the rope.

**Various Physical Properties.** One key feature that distinguishes the proposed ContPhy dataset from existing benchmarks like CLEVRER (Yi et al., 2020) and Physion (Bear et al.) is that our proposed benchmark varies the values used for physical parameters like masses, density, elasticity, and stretchiness. Such variation in the values will lead the dynamics to different future states. As such we can generate a richer dataset that covers a larger variance in object physical parameters. For example, the density of the liquid in Fig. 1 (a) will decide where the liquid stays, and the stretchiness of the cloth in Fig. 1 (b) will determine how the objects will contact with the cloth.

**Simulation Setup.** In the context of video and annotation generation, we follow a bottom-up approach, involving the following sequential steps: a) Sampling: we begin by randomly selecting scene layouts, camera parameters, and initial conditions to create a diverse set of scenarios. b) Initialization: objects are placed within the scene based on the sampled layout. c) Pre-simulation: a preliminary simulation is conducted to evaluate whether the obtained simulation results align with the expected data distribution. d) Rendering: high-quality videos are generated by configuring camera settings. e) Post-simulation: multiple simulations are carried out under varying conditions, and the simulation outputs are recorded. f) Output: sensor data and annotation information are produced, encompassing not only the original video but also semantic segmentation, bounding boxes, point clouds, mesh surfaces, collision detection data, camera parameters, raw sampling data, and other simulation results required for question generation. We provide more details on this procedure in the appendix. In terms of detailed settings, four diverse scenarios were simulated.

**Liquid Dynamics.** As shown in Fig. 2 A, we have designed a device that bears a resemblance to a liquid hourglass. In this device, various liquids of different densities, each represented by distinct colors, are released from corresponding emitters situated at the uppermost part of the apparatus. Under the influence of gravity, these liquids descend and traverse a series of fixed ramps (resembling sticks) positioned in the central region of the device. This arrangement causes alterations in their flow direction. Ultimately, liquids are funneled into containers at the bottom, each container designated by a unique color. This process highlights distinctive behaviors arising from the interaction of multiple fluids, attributable to their significantly varied densities. Our research is oriented towards formulating inquiries pertaining to physical properties of these liquids and dynamic trajectories they exhibit.

**Cloths Manipulation.** As depicted in Fig. 2 B, a small table hosts an assortment of objects, including pillars and plates of varying sizes, colors, and masses. Two square pieces of cloth, each possessing distinct stretching, bending characteristics, and frictional properties, are gripped at one edge and moved forward to cover these objects, causing possible collision events. Cloths are then promptly released. The fabric obstructs the view of the objects but also delineates their shapes through its deformable surface. Objects may topple over if they exceed a certain height or have low mass, resulting in observable changes in the fabric's dynamic 3D surface geometry. This scenario serves as a test for a model's capacity to discern the physical attributes of the fabrics and to predict the spatial behavior of the concealed objects in dynamic situations.

**Rope Pulley System.** As illustrated in Fig. 2 C, an array of pulleys, including both movable and fixed types, along with anchor points, is arranged on a wall. Ropes are configured with their ends connected to pulleys, loads, or anchor points, and can be wound around the pulleys. These loads possess varying masses, interacting with other forces in the system, leading to the emergence of distinct motion patterns. The primary objective of the model is to identify the tension distributions within this elementary rope system. Additionally, it is tasked with recognizing potential correlations or constraints among objects in motion, such as the coordinated movement of loads and the rotation of pulleys on a single rope. Moreover, the model is expected to infer numerical relationships between the loads' masses and predict whether the ropes will detach from or remain attached to certain pulleys.

**Soft Ball Dynamics.** As depicted in Fig. 2 D, a playground contains obstacles of different color, and pose, along with pits randomly arranged within. Soft balls with varying deformation resistance or plasticity yield are launched randomly within the space, with varying initial positions. These balls undergo a sequence of dynamic movements, including bouncing and permanent deformation. Ultimately, some may collide with obstacles and fall into pits. This experimental scenario serves as a

test to determine whether the model can accurately discern the elasticity and plasticity properties of the soft bodies. It also assesses the model's ability to make dynamic predictions and inferences based on these attributes during interactions between objects.

## 3.2 QUESTION GENERATOR

**Generation Steps.** We develop a question generator to generate question-answering pairs associated with the videos through the following steps: a) Template Design: create a range of question and option templates for generation (See table A.2 in the appendix). b) Sampling: retrieve the simulation results, combine the properties of the objects in the scene with predefined templates, and sample questions and options accordingly. Correct answers are determined based on the simulation outcomes. Unique identification and description of target objects are accomplished using visual attributes such as color, shape, orientation, and mobility. c) Re-Sampling: ensure a balanced distribution of answers among the options to prevent answer bias toward any particular choice.

**Overview.** We have categorized our questions into two major groups: **Physical Property Questions** and **Dynamics Questions**. Figure 2 shows all of the question types present in ContPhy for each of the four scenarios. Sample templates are provided in Appendix A.2.

**Physical Property Questions.** We formulated a set of physical property questions across four distinct scenarios. These questions inquire about visible physical properties of objects, such as colors, shapes, and existences, which can be discerned from static video frames. Additionally, we pose questions about physical attributes, including mass, density, and deformability, which can only be answered by observing various object dynamics and interactions. These questions primarily revolve around factual features that can be perceptive and answered with a brief phrase response. Models are expected to deduce these physical properties based on input video data, which requires models to possess a foundational understanding of fundamental physical principles.

**Dynamics Questions.** Regarding dynamic questions, we explored various scenarios involving the behavior of objects. Dynamic questions can be further categorized into three distinct types: counterfactual, goal-driven, and predictive. These questions encompass potential outcomes when initial conditions change, strategies for achieving specific objectives and inquiries about future events. In the cloth scenario, we only designed predictive questions encouraging the model to anticipate outcomes not directly visible under the cloth cover. In the rope scenario, we only have exclusively formulated counterfactual and goal-driven questions, aimed at motivating models to simulate hypotheses regarding the soft-body-driven mechanical transmission scenario. For the remaining two scenarios, fluid and ball, we have designed questions encompassing all three types, with the expectation that models will acquire a comprehensive understanding of these scenarios through the diverse nature of the question templates. To enhance the cognitive challenge, we have structured these questions as multiple-choice, featuring more than two but fewer than five answer choices. Models are tasked with providing binary true-false predictions by concatenating their response with the corresponding question-choice pair.

## 3.3 STATISTICS AND POST-PROCESSING

In our dataset, we have generated a substantial volume of videos, questions, physical parameters, and objects. To provide a more detailed breakdown:

**Video Data.** We categorize videos by scenario. Each scenario contains 500 videos of fixed lengths: 250 frames for fluids, 150 for ropes, 145 for cloths, and 120 for balls. Given the diverse responses in the VQA generation phase, we employed randomization for several configuration parameters during the simulation initialization. Beyond general scene arrangements like camera, lighting, and backgrounds, unique configurations pertain to each scenario:

a) Fluids: Fluid density factors into multi-fluid interactions. Striving for diverse results, the number of fluid emitters and containers, the positions, poses, scales of obstructive sticks, and object colors are randomized. Fluid densities, chosen from a preset pool, should ensure discernible stratification in fluid interactions.

b) Ropes: The rope-pulley system layout, rope link lists, and entanglement methods are pre-set to allow varied connections between adjacent objects. Filtering steps identify simulations that provide

diverse and aesthetically pleasing configurations. Attributes such as color, shape, load mass, load movability for loads, ropes, fixed endpoints, and pulleys are randomized prior to simulation.

c) Cloths: Parameters like stretching compliance, bending compliance, and friction rate are drawn from a predetermined pool, ensuring cloth dynamic differences discernible to humans. Other items, such as pillars and plates, undergo random scaling and positioning. Cloth movement speeds and paths vary, aiming for diverse collision outcomes. Rigid object masses are also randomized to diversify collision event predictability.

d) Balls: Deformation resistance and plasticity yields are sourced from a set value range to highlight differing properties. Floating wall positions and poses are constrained to specific zones to intensify collision events in videos, leading to varied outcomes during and post-video.

**Question Data.** In accordance with the video content, we have formulated a substantial quantity of questions. Each video has one property question and two dynamics questions, except for rope scenario which has two property-related questions and two dynamics questions. We generated a total of 2,000 questions related to the rope scenario and 1,500 questions related to other scenarios, respectively.

Consequently, our dataset comprises a total of 6,500 questions drawn from 2,000 videos. We have partitioned the dataset into three subsets: 50% for training, 20% for validation, and 30% for testing. More precisely, the training set consists of 3,250 questions, the validation set comprises 1,300 questions, and the testing set encompasses 1,950 questions. Through the whole dataset, 20% are counterfactual, 11% are goal-driven, 22% are predictive, and the remaining 46% pertain to physical property questions of various kinds. The detailed distribution of each question type within the rope scenario is visualized in Figure 3, while Table 4 provides templates for each rope-related question type. Further information about question types of other scenarios is available in Appendix A.2.1.

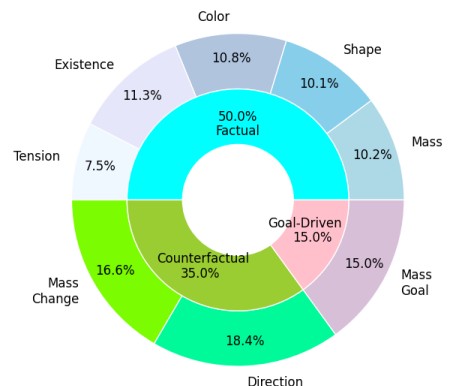

Figure 3: Question distribution of rope scenario.

## 4 EXPERIMENTS

In this section, we perform evaluation and analysis on various baseline models for our ContPhy dataset for video reasoning. In accordance with the standard evaluation protocols that have been adopted in previous works such as CLEVRER, we consider each physical property question as a classification task among all possible answers. Each dynamic question is treated as a binary classification task for each question-choice pair, as dynamic questions are always multiple-choice questions. For dynamic questions, we report the accuracy for each option and per question. A question is correct only if all choices in this multiple choice question are correctly answered.

### 4.1 EXPERIMENTAL SETUP

**Visual-Blind Models.** This family of models include several baselines that only rely on question-only input, to help us analyze language biases in ContPhy. **Random** chooses at random a possible answer, or randomly selects between true-false binary answer pairs for every multiple choice questions. **Frequent** selects the most frequent answer based on the question type. **Blind-LSTM** with language-only input utilizes an LSTM network (Hochreiter & Schmidhuber, 1997) to encode the question and predict the answer.

**Visual Models.** These models incorporate both visual and language representations for answering questions about physical events in videos. **CNN-LSTM** extracts video features via ResNet-50 convolutional neural network (CNN) (lec, 1998; He et al., 2016) on 25 sampled frames of videos and average them over time as the visual input. We concatenate this visual input with the question embedding from the last hidden state of LSTM to predict answers. **HCRN** (Le et al., 2020) uses

| Methods | Rope | | | | | Fluid | | | | | | |
|---|---|---|---|---|---|---|---|---|---|---|---|---|
| | Prop. | Counter. opt. | Counter. ques. | Goal. opt. | Goal. ques. | Prop. | Counter. opt. | Counter. ques. | Goal. opt. | Goal. ques. | Pred. opt. | Pred. ques. |
| Random | 30.0 | 51.3 | 14.7 | 55.2 | 4.5 | 33.3 | 52.9 | 6.0 | 59.9 | 7.5 | 53.8 | 4.8 |
| Frequent | 53.3 | 51.6 | 19.0 | 49.7 | 11.2 | 52.7 | 57.9 | 17.2 | 63.1 | 36.3 | 50.1 | 12.5 |
| Blind-LSTM | 54.7 | 74.0 | 46.0 | 47.4 | 7.9 | 49.3 | 56.1 | 7.8 | 57.3 | 22.5 | 51.4 | 12.5 |
| CNN-LSTM | 52.7 | 74.0 | 45.0 | 51.2 | 6.7 | 54.0 | 55.0 | 8.6 | 57.3 | 22.5 | 51.4 | 12.5 |
| MAC | 53.3 | 74.2 | 39.8 | 50.3 | 6.7 | 30.0 | 56.5 | 6.9 | 51.2 | 17.5 | 53.5 | 12.5 |
| HCRN | 51.7 | 74.3 | 48.1 | 56.0 | 2.3 | 52.7 | 52.6 | 4.3 | 67.7 | 41.3 | 50.6 | 1.9 |
| ALPRO | 60.7 | 76.2 | 50.7 | 46.2 | 1.1 | 48.0 | 56.8 | 6.0 | 62.7 | 32.5 | 53.8 | 12.7 |
| Human | 84.7 | 90.2 | 75.0 | 91.9 | 84.0 | 75.8 | 82.5 | 60.6 | 75.0 | 64.3 | 73.9 | 42.9 |

| Methods | Cloth | | | Ball | | | | | | |
|---|---|---|---|---|---|---|---|---|---|---|
| | Prop. | Pred. opt. | Pred. ques. | Prop. | Counter. opt. | Counter. ques. | Goal. opt. | Goal. ques. | Pred. opt. | Pred. ques. |
| Random | 46.7 | 52.2 | 46.0 | 53.5 | 53.6 | 30.4 | 55.9 | 30.2 | 50.6 | 25.9 |
| Frequent | 41.3 | 61.7 | 56.7 | 52.0 | 65.8 | 48.7 | 52.1 | 38.5 | 67.8 | 51.7 |
| Blind-LSTM | 56.7 | 55.2 | 42.3 | 45.3 | 66.7 | 43.4 | 53.3 | 16.7 | 68.9 | 45.5 |
| CNN-LSTM | 46.7 | 67.5 | 57.3 | 54.7 | 64.2 | 41.8 | 54.1 | 20.0 | 67.4 | 45.5 |
| MAC | 59.3 | 57.9 | 50.7 | 48.0 | 66.1 | 3.3 | 58.1 | 18.9 | 64.4 | 46.6 |
| HCRN | 52.0 | 62.0 | 56.3 | 43.3 | 65.3 | 28.7 | 57.0 | 38.9 | 61.7 | 1.1 |
| ALPRO | 48.0 | 68.8 | 57.3 | 48.0 | 63.9 | 40.2 | 56.3 | 4.4 | 65.2 | 3.4 |
| Human | 81.4 | 79.6 | 77.3 | 76.9 | 93.9 | 90.9 | 89.7 | 84.6 | 72.5 | 58.8 |

Table 2: Physical reasoning on ContPhy. We list all question families, **Prop**erty, **Counter**factual, **Goal**-driven and **Pred**ictive questions. Accuracy is reported with per option and per question.

conditional relational networks to learn relations hierarchically in the video, as well as the questions. **MAC** (Hudson & Manning, 2018) has competitive results on previous datasets, which uses co-attention mechanism to model both textual and visual information. **ALPRO** (Li et al., 2022a) is a popular videoQA model pre-trained on video-text feature alignment. After stages of contrastive learning at instance level, and prompting entity fine-grained modeling between patch and text entity, ALPRO achieved state-of-the-art results on several video multi-modal datasets. We fine-tune on our dataset based on the official pre-trained checkpoint.

## 4.2 EVALUATION OF PHYSICAL INFERENCE

We summarize the performance of all baselines in Table 2. The results show that different models exhibit distinct performances across different scenarios, even on different question types within a single scene. This indicates that our ContPhy benchmark encompasses a wide range of reasoning taks, making it a valuable tool for evaluating the limitations of visual models.

**Physical Property.** Physical property questions in ContPhy focus on both the fundamental content in the video, as well as properties governed by physical rules. This implies that models should not only recognize content simply but also understand the video and possess physical knowledge, putting forward a new challenge to models. None of the models successfully addressed all of these types of questions. Some baselines perform worse than language-only models in certain scenarios, indicating a failure to correctly understand physical properties and the importance of our dataset. Most of the baseline frameworks are not specifically designed for physical properties, which accounts for the poor performance. All the baseline models struggle to achieve decent performance on physical property questions, except ALPRO, which is in the rope scenario and maintains competitive results in other scenarios, showing the advantages of large-scale video-text pre-training and alignment.

**Dynamics.** Dynamics questions including counterfactual, goal-driven, and predictive, pose another challenge that further enhanced reasoning from dynamic video sequences. These questions require models to focus not only on visual perception but also on predicting unseen information. Models are required to fully comprehend the video and make inferences or predictions based on the questions.

All dynamics questions are multiple-choice, although the number of choices varies among different datasets. This also explains the different accuracy per question on Random and Frequent. We find that ALPRO achieves better performance on counterfactual reasoning, proving the superiority of its pre-training framework again. HCRN has advantages in goal-driven question reasoning, but fails in predictive questions. We postulate that a possible reason could be that its hierarchical network mechanism equips the model with the ability to explore unseen scenarios under goal-driven conditions but not for predictive questions. Other baselines based on traditional neural networks have difficulty understanding the physical scenarios and in capturing the physical laws from videos and question-answer pairs. Thus, they perform worse than their previous performance on our benchmark.

**Scenario Analysis.** We observe that different models exhibit significant variability in their performance across different scenarios. Regarding physical properties, CNN-LSTM performs well on fluid and ball, MAC excels with cloth and ALPRO demonstrates strong performance in the rope scene. For dynamics questions, ALPRO answers well in both cloth and rope scenario. A possible explanation is that both cloth and rope scenarios share some similarities as they both exhibit mechanical event such as collision, motion, and rotation led by soft objects as opposed to the fluid and ball scenario. Another reason is the fewer question types in the rope and cloth scenario than those in the fluid and ball scenarios. Specifically, the rope scenario has counterfactual and goal-driven, and the cloth scenario has predictive. Conversely, in the fluid and ball scenarios, we incorporated all four problem types, thereby making situations much more complicated. To effectively address these scenarios, models must tackle four distinct question types, each focusing different aspects of physical dynamics. Consequently, no baseline models can gain an absolute advantage in these scenarios. This indicates that our four proposed question types well evaluate different dimensions of physical reasoning, making the fluid and ball scenarios particularly challenging for AI models. In addition, visual models marginally outperform language-only models, suggesting that existing models struggle to comprehend complex soft-body visual information and interaction with other rigid bodies.

**Human Performance.** We randomly sampled some video-question pairs from the test set in order to assess human ability to comprehend the physical properties and dynamics events presented in both video and textual descriptions. To evaluate human performance on ContPhy, 16 people participated in the study. Participants were required to have fundamental English reading skills and basic physical knowledge background. First, each participant was asked to select a scenario randomly, after which they presented with distinct video-question pairs. Participants were instructed to answer with a phrase when presented with physical property questions, while for dynamics questions they were required to provide a binary true-false response from available choices. We obtained 460 valid human answers encompassing all scenarios and question types within ContPhy. Human Performance is presented in Table 1 and we can observe that it beats machine models in all scenarios. This shows the fundamental ability and strength of humans to perform visual reasoning and inference from videos.

**Evaluation Conclusion.** The strong human results demonstrate that humans maintain a strong capacity to comprehend both videos and questions, make physical property inference from given videos, and predict and reason counterfactual hypothesis concerning unseen information. Machine models results show that even state-of-the-art models struggle with answering these physical questions. This indicates that our dataset poses a significant challenge for vision-language models to achieve similar basic physical video understanding ability with human beings. Our dataset is crucial for assessing and advancing video-text understanding as well as physical inference capacity.

## 5 CONCLUSION

We introduced the Continuum Physical Dataset (ContPhy), a pioneering benchmark for assessing machine models in physical reasoning of the continuum, especially for soft bodies and fluids in the continuum. This benchmark broadens the scope by covering various physical property inferences for soft bodies across dynamic contexts and predicting their dynamics. Our dataset has enabled the development of AI models with human-like reasoning abilities, comprehending both visual attributes and complex physical properties of objects while solving problems. Despite progress, our evaluation of AI models revealed an ongoing challenge: they struggle to perform well on our benchmark, highlighting their limited physical commonsense for the continuum, especially soft bodies and fluids. We foresee the ContPhy driving progress in AI perception and reasoning, bridging the gap between human and machine intelligence in the physical world.

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

# A APPENDIX

This appendix aims to provide more details and qualitative examples to support our claim on the main paper.

## A.1 DATASET DETAILS

For each simulation trial, we produce two primary sets of data: sensor output and semantic annotation. The sensor output provides a comprehensive 4D state description of objects at various levels. In contrast, the semantic annotation contains pre-processed data designed to facilitate the question generation phase.

### A.1.1 SENSOR DATA STRUCTURE.

Within the simulation pipeline, we produce sensor data across multiple modalities listed in Fig. 4, including RGB-rendered images in Full HD (1920x1080) resolution, object-level data (encompassing bounding boxes, segmentations, positions, rotations, and scales), point-level data (comprising meshes and particles), and event-level data (detailing collision or touch events). The generated meshes illustrate the sampled surface shapes of both rigid and soft objects, prepared for subsequent voxelization. Unlike the other two scenarios, the fluid and rope scenarios necessitate the re-sampling of meshes in every individual frame. This results in temporal independence for the vertices. Yet, within this context, particle outputs signify tracked points on the objects, preserving correlations between successive frames. Given that voxel data (which is temporally invariant) is derived from the voxelization of meshes, dataset offers both temporally correlated and independent 4D data.

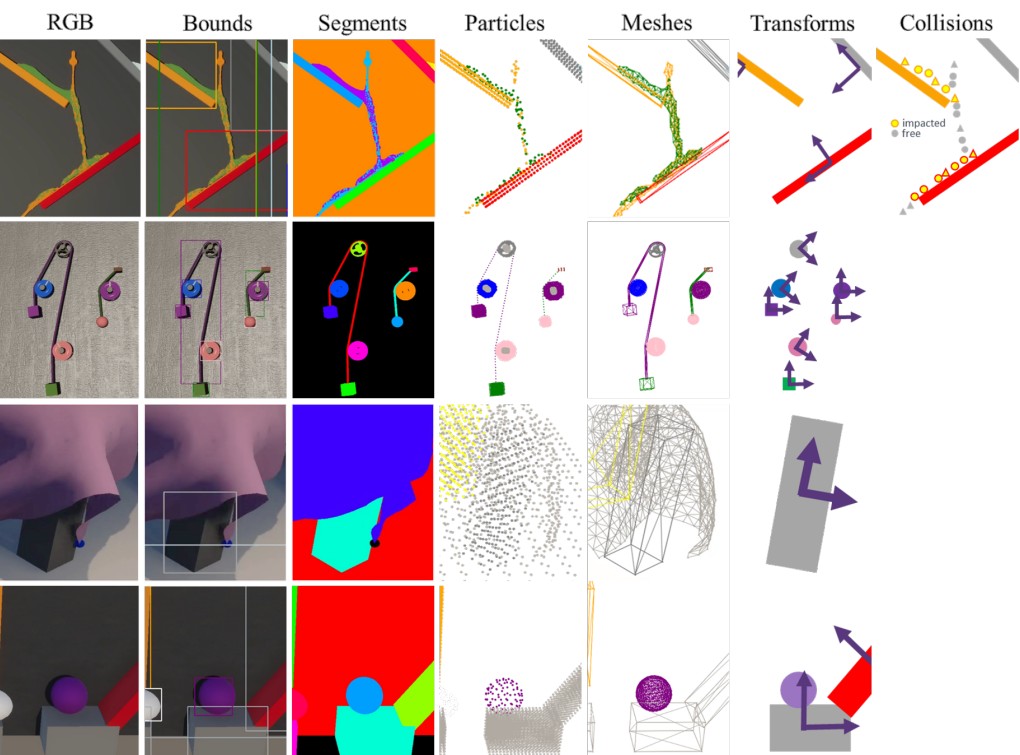

Figure 4: Sensor data outputs are multimodal, depicting the 4D states of objects across various levels, ranging from object-level, point-level to event-level.

### A.1.2 ANNOTATION DATA STRUCTURE.

For each scenario, we produce comprehensive annotation data that includes camera extrinsics/intrinsics, sampled parameters, and properties of the sampled objects and layouts. Additionally,

post-processed simulation data from both the pre-simulation and post-simulation stages are documented. To be specific:

**Fluid.** Object details such as name, color, and transforms are stored. For fluid objects, properties like densities, viscosity, surface tension, and emitted positions are added. Particle statistics in each container, collision statistics on each stick, and collision paths for each particle are recorded for both pre- and post-simulation stages, and are meticulously categorized by fluid types.

**Rope.** Fundamental elements of each pulley group, such as pulley, rope, fixed endpoint, cube, and sphere, are outlined at both the individual rope and group levels. A group refers to a collection of objects with interdependent mechanics, like two sets of objects on ropes connected to a specific movable pulley. Initial properties such as mass, color, shape, mobility, pose, and subsequent simulation results like motion direction of movable objects and tension in rope segments are annotated.

**Cloth.** Sampled cloth properties—stretching compliance, bending compliance, and friction level—are provided. Basic properties of each rigid object and their simulation results, which include object-cloth and object-object collision events, contact relationships, and tension values in the cloth's final frame, are stored.

**Ball.** The framework documents sampled properties of all rigid bodies and soft balls. For plastoelastic balls, simulation results, including the pits they settle into, are captured.

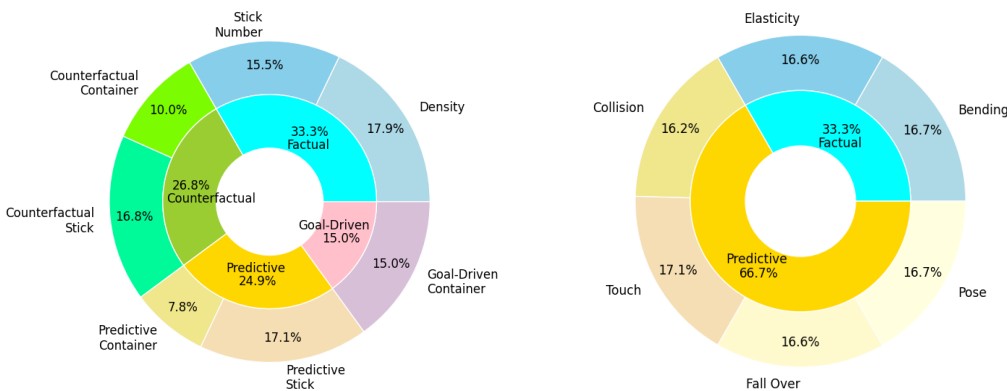

Figure 5: Question distribution of fluid scenario.    Figure 6: Question distribution of cloth scenario.

## A.2 QUESTION DETAILS.

### A.2.1 QUESTION DISTRIBUTION

In this section, we visualize the distribution of question types in fluid, cloth and ball scenarios in Figure 5, 6 and 7. Rope scenarios results are in Figure 3. We balance the number of each question type.

### A.2.2 QUESTION TEMPLATES

In this section, we show all of our question templates from four scenarios in Table 4, 5, 6, and 7. We define all the symbols we use in Table 8. When generating questions using templates and symbols, we balance the distribution and frequency of each symbol and answer to avoid language bias.

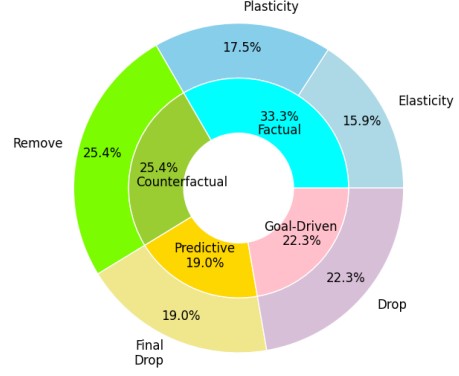

Figure 7: Question distribution of ball scenario.

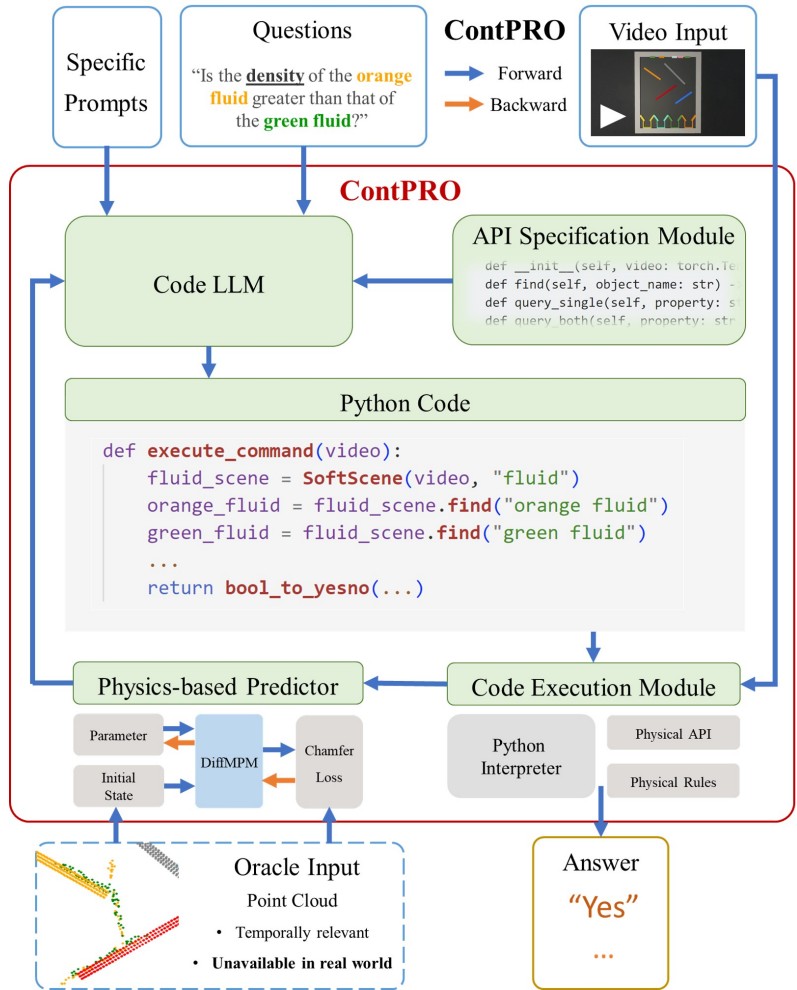

Figure 8: Model architecture for **Cont**inuum **P**hysical **R**easoning **O**racle Model, **ContPRO**. With specific prompts and questions, code LLM will generate Python code for physical reasoning. Differentiable Physics-based Predictor will predict dynamics in the scenario with 3D point cloud input. Code Execution Module utilizes physical APIs as well as physical rules to output answers directly.

## A.3   CONTPRO: CONTINUUM PHYSICAL REASONING ORACLE MODEL

We introduce **Cont**inuum **P**hysical **R**easoning **O**racle Model, **ContPRO**, which marries physical-based dynamics models with the recent large language models which enjoy the advantages of both models, precise dynamic predictions, and interpretable reasoning. We discuss our model framework in A.3.1, as well as the large language model backbone, the visual perception model, and the physical simulation model. We also present the representative results and analysis in A.3.2.

### A.3.1   MODELS

Inspired by prior works (Yi et al., 2018; Chen et al., 2022; Yi et al., 2020), we propose an oracle neural-symbolic framework named ContPRO. As the rapid development of large language models and their applications, we replace previous rule-based neural executor with the newest neuro-symbolic large language models. Furthermore, our oracle model is equipped with a state-of-the-art differentiable physical simulatorand the popular visual perception model, MASK R-CNN (He et al., 2017). This physical simulator uses a point cloud as initial input, which is hard to perceive directly from videos. That accounts for the reason that we call it oracle model.

**Large Language Model Backbone.** Recent years, we have witnessed a significant breakthrough in large language models. Works (Surís et al., 2023; Gupta & Kembhavi, 2023) introduce neural-

symbolic style into large language models, proposing a powerful method for solving complex visual queries and semantic information within languages. ChatGPT and its family (Ouyang et al., 2022; OpenAI, 2023; Brown et al., 2020) have absolute advantages in understanding complex semantic information in our generated questions, saving us a lot of trouble and maintaining great parse accuracy simultaneously. With the aid of large language models, semantic information and physical concept can be precisely parsed out without manually develop a rule-based question parser and question programs.

Similar to ViperGPT (Surís et al., 2023), we utilize large language models, specifically ChatGPT, as the backbone of our proposed ContPRO. We further develop a set of dynamics modules and visual perception modules serving as APIs for code generation model. With the provided API access and a pre-defined physical reasoning prompt, we leverage ChatGPT to generate Python code that can be directly executed and interpreted. This method bridge the gap between language comprehension and physical concept understanding, enabling us to directly evaluate the continuum physical reasoning abilities of AI models, without concerns related to different expressions of question sentences.

**Video Perception.** We adopt MASK R-CNN (He et al., 2017) as our video perception module. This module aids ContPRO to detect and ground objects in videos, bridging the gap between linguistic descriptions of objects and their physical properties. Given an input video, this module will detect objects in each frame, addressing simple property questions directly, such as object counting, object existence and color recognition, with very high accuracy. Also its ability to locate objects through bounding boxes indirectly plays a role in other questions. For example, in the Density question of the fluid scenario, fluids' densities are comparable only if they are in the same container. This location ability enables us to discern whether they are in the same container, i.e., their densities are comparable.

**Physical Simulation.** Previous methods (Chen et al., 2022; Bear et al.; Tung et al., 2023), mainly adopt DPI-Net as physical simulator, which is a 3D particle-based graph neural network (Li et al., 2018). DPI-Net performs well in mass (Tung et al., 2023), while falls short in fluid dynamics prediction. Therefore, we utilize a differentiable MPM physics simulation module DiffMPM (Hu et al., 2019) to provide a primary solution to the system identification of the fluid scenario. We also borrowed the simulation framework partly from PAC-NeRF (Li et al., 2023).

To show the details, we illustrate the pipeline solving fluid system identification as Fig. 8. In the physical system identification phase, we firstly initialize the density of each kind of fluid with a fixed value. We use the fluid point cloud as the initial input. For each iteration, we repeatedly feed initial points forward in the differential physics simulation pipeline, which is rewritten to fit in 2D vector computation for better performance. At the meantime, the loss is constructed as the chamfer loss between the groups of fluid points and the predicted points. Then the gradients are fed back to the optimization of each physical parameters, i.e. densities here. In the prediction phase, the fluids' trajectories are predicted, with the simulation including the fluid-rigidbody collision and fluid-fluid interaction (i.e. liquid stratification). Collision events and layer information are recorded and output for the downstream QA tasks. Likewise, in the counterfactual prediction phase, we repeatedly change the initial conditions and rerun the simulation.

### A.3.2 REPRESENTATIVE RESULTS ANALYSIS

The physical simulation method (Hu et al., 2019; Li et al., 2023) in oracle model exhibits powerful performance that can precisely simulate the continuum dynamics, especially soft bodies, while it requires scenario-specific manipulation of hyperparameters. Therefore, we present representative results in the fluid scenarios below, which encompasses all four question types, and compare them with multiple baseline models. We report the representative results in Table 3, which reveals that our ContPRO successfully outperforms all questions in the fluid scenario. These results indicate that ContPRO is capable of understanding both semantic information in questions and the continuum physical concept well.

**Physical Property.** Specifically, in property question, ContPRO attains a remarkable accuracy of 78.0% in overall property question. The property question consists of two distinct tasks: Stick Number, which represents a relatively simple task, and Density, which is more challenging. In Stick Number, ContPRO achieves 96.9%, as this task can be well solved by visual perception models solely, only requiring Mask R-CNN to correctly recognize objects in videos. In Density, ContPRO

| Methods | Property | Counterfactual | | Goal-Driven | | Predictive | |
|---|---|---|---|---|---|---|---|
| | | per opt. | per per ques. | per opt. | per ques. | per opt. | per ques. |
| Random | 33.3 | 52.9 | 6.0 | 59.9 | 7.5 | 53.8 | 4.8 |
| Frequent | 52.7 | 57.9 | 17.2 | 63.1 | 36.3 | 50.1 | 12.5 |
| Blind-LSTM | 49.3 | 56.1 | 7.8 | 57.3 | 22.5 | 51.4 | 12.5 |
| CNN-LSTM | 54.0 | 55.0 | 8.6 | 57.3 | 22.5 | 51.4 | 12.5 |
| MAC | 30.0 | 56.5 | 6.9 | 51.2 | 17.5 | 53.5 | 12.5 |
| HCRN | 52.7 | 52.6 | 4.3 | 67.7 | 41.3 | 50.6 | 1.9 |
| ALPRO | 48.0 | 56.8 | 6.0 | 62.7 | 32.5 | 53.8 | 12.7 |
| **ContPRO-Oracle** | **78.0** | **75.7** | **36.2** | **77.3** | **60.0** | **90.1** | **68.3** |

Table 3: Physical reasoning in the fluid scenario on ContPhy. We list all question families, including Property, Counterfactual, Goal-driven and Predictive questions. Accuracy is reported with per option and per question.

achieves 63.5% accuracy, which is a more intricate challenge involving density comparisons of two fluids, requiring a linkage of multiple modules. This task presents a more intricate challenge and requires a linkage of multiple modules. Firstly, the process begins with language models parsing the question and generating Python code. Subsequently, the visual perception module recognizes and locates these fluids in the video by their names, and outputs bounding boxes of two fluids. A specialized detection function then determines if the fluids share the same container by analyzing their location and overlap, which is a prerequisite for comparability of two fluids. Finally, if the fluids are comparable, the physical simulator steps in to infer the relative densities. In general, the high accuracy achieved in the Stick Number task significantly boosts the overall property question accuracy, positioning ContPRO well above other models in this question.

**Dynamics.** In the dynamics question set, ContPRO notably excels, surpassing other baseline models across all categories. This superior performance indicates that our proposed model can accurately simulate the continuum physical scenarios via a differentiable physical dynamic simulator. This accurate simulation enhances the model's ability to reason and predict dynamics in subsequent stages. Among the three types of dynamics questions, our oracle model particularly outperforms others in predictive questions, where it achieves 68.3% accuracy per question. This discrepancy may be attributed to the training objectives of MPM physical simulator in ContPRO. In MPM prediction phase, simulator focuses on predicting trajectories and dynamics. Notably, this simulator-based method requires 3D point cloud information which cannot be perceived from 2D videos. Thus we consider it as an oracle model baseline.

## A.4 THE CONTINUUM: LIQUIDS, SOFT BODIES, RIGID BODIES AND ARTICULATED BODIES

In this section, we consider the physical concept of the continuum. Previously, physical datasets mainly focused on simple visual primitives of rigid bodies, such as cubes and spheres. In our ContPhy, we extend this success to a broader concept, the continuum. The continuum encompasses various bodies such as liquids, soft materials (*e.g.*, soft balls, cloth, and ropes), rigid bodies (*e.g.*, cubes, pillar, plate and spheres), and articulated bodies (*e.g.*, pulleys). We consciously include both physical dynamics reasoning (*e.g.*, interactions between fluids, soft bodies and rigid bodies), and physical parameter or concept reasoning (*e.g.*, density for fluids; tension, elasticity for soft bodies; mass for rigid bodies).

For instance, our rope and pulley scenario involves elements of rope, rigid bodies, and articulated bodies; the fluid scenario includes liquids; the cloth scenario covers both cloth and rigid bodies; and the ball scenario focuses on soft balls. This extensive coverage ensures that our dataset provides a comprehensive understanding of the interactions and couplings within these various types of continua, capturing the complexity and diversity of real-world physical phenomena.

In our paper, we focus predominantly on fluids and soft bodies, which is often overlooked in previous works. However, our dataset comprehensively encompasses rigid bodies in all scenarios and articulated bodies (*e.g.*, in the rope scenario). This inclusion leads to our utilization of the continuum concept, enhancing the breadth and relevance of our study.

| Class | Type | Template and Example |
|---|---|---|
| Shape | Factual | **Q** How many _SHP_ _OBJ_s are there in the video?
***E.g.*** How many solid pulleys are there in the video? |
| Color | Factual | **Q** How many _CLR_ objects are there in the video?
***E.g.*** How many blue objects are there in the video? |
| Existence | Factual | **Q** Is there any _CLR_ _OBJ_ in the video?
***E.g.*** Is there any blue cube in the video? |
| Mass | Factual | **Q** Is the mass of the _CLR_ _OBJ_ _CMP_ _FAC_ that of the _CLR_ _OBJ_?
***E.g.*** Is the mass of the blue sphere greater than half that of the green cube? |
| Tension | Factual | **Q** Is the tension of the _CLR_ rope _CMP_ _FAC_ that of the _CLR_ rope?
***E.g.*** Is the tension of the blue rope greater than half that of the green rope? |
| Mass Change | Counterfactual | **Q** If the _CLR_ _OBJ_ were heavier, what would happen?
***E.g.*** If the blue sphere were heavier, what would happen? |
| Direction | Counterfactual | **Q** If the _CLR_ _OBJ_ were much heavier, which direction would the _CLR_ _OBJ_ move?
***E.g.*** If the blue cube were much heavier, which direction would the brown sphere move? |
| Mass Goal | Goal-Driven | **Q** If we want the _CLR_ _OBJ_ to _ENT_, what can we do?
***E.g.*** If we want the yellow cube to move up, what can we do? |

Table 4: Question templates and examples in Rope.

| Class | Type | Template and Example |
|---|---|---|
| Density | Factual | **Q** Is the fluid density of the _CLR_ fluid _CMP_ that of the _CLR_ fluid?
**E.g.** Is the fluid density of the pink fluid greater than that of the light blue fluid? |
| Stick Number | Factual | **Q** How many sticks are there in the video? |
| Pass | Predictive | **Q** Which stick will the fluid from the other _CLR_ emitter pass?
**E.g.** Which stick will the fluid from the other blue emitter pass? |
| Container | Predictive | **Q** Which container will fluid from the other _CLR_ emitter flow into?
**E.g.** Which container will fluid from the other blue emitter flow into? |
| Pass | Counterfactual | **Q** If _CLR_ stick were removed, which stick would _CLR_ fluid pass?
**E.g.** If brown stick were removed, which stick would pink fluid pass? |
| Container | Counterfactual | **Q** If _CLR_ stick were removed, which container would _CLR_ fluid flow into?
**E.g.** If brown stick were removed, which container would pink fluid flow into? |
| Container | Goal-Driven | **Q** What can we do to let most of the _CLR_ fluid enter _CLR_ container?
**E.g.** What can we do to let most of the pink fluid enter gray container? |

Table 5: Question templates and examples in Fluid.

| Class | Type | Template and Example |
|---|---|---|
| Elasticity | Factual | **Q** Is the elasticity of the _POS_ cloth much _CMP_ that of the other?
**E.g.** Is the elasticity of the left cloth much greater than that of the other? |
| Bending | Factual | **Q** Is the _POS_ cloth much _CMP_ to bend or have wrinkles than the other?
**E.g.** Is the right cloth much harder to bend or have wrinkles than the other? |
| Fall Over | Predictive | **Q** Does the _CLR_ _OBJ_ fall over?
**E.g.** Does the green plate fall over? |
| Collision | Predictive | **Q** Does the _CLR_ _OBJ_ collide with the _CLR_ _OBJ_?
**E.g.** Does the green plate collide with the gray pillar? |
| Touch | Predictive | **Q** Is the _CLR_ _OBJ_ finally in touch with the _CLR_ _OBJ_?
**E.g.** Is the green plate finally in touch with the gray pillar? |
| Pose | Predictive | **Q** Which phrase below can best describe the final pose of the _CLR_ _OBJ_?
**E.g.** Which phrase below can best describe the final pose of the green plate? |

Table 6: Question templates and examples in Cloth.

| Class | Type | Template and Example |
|---|---|---|
| Elasticity | Factual | **Q** Is the elasticity of the _CLR_ ball much _CMP_ the _CLR_ ball? 
 **E.g.** Is the elasticity of the brown ball much greater than the purple ball? |
| Plasticity | Factual | **Q** Is the plasticity of the _CLR_ ball much _CMP_ the _CLR_ ball? 
 **E.g.** Is the plasticity of the brown ball much greater than the purple ball? |
| Final Drop | Predictive | **Q** Will the _CLR_ ball finally drop into the _POS_ pit? 
 **E.g.** Will the brown ball finally drop into the left pit? |
| Remove | Counterfactual | **Q** If we removed the _CLR_ floating wall and other balls, which pit would the _CLR_ ball drop into? 
 **E.g.** If we removed the yellow floating wall and other balls, which pit would the brown ball drop into? |
| Drop | Goal-Driven | **Q** What can we do to make the _CLR_ ball drop into the _POS_ pit? 
 **E.g.** What can we do to make the pink ball drop into the right pit? |

Table 7: Question templates and examples in Ball.

| Symbol | Explatations |
|---|---|
| _CLR_ | blue, black, brown , cyan , gray , green , orange , pink , purple red , yellow, light blue, white |
| _SHP_ | solid, hollow |
| _OBJ_ | plate, pillar, cube, sphere, pulley, rope |
| _CMP_ | greater than, less than, harder, easier, equal to |
| _FAC_ | twice of, half of |
| _POS_ | left, right |
| _ENT_ | move up, move down, rotate clockwise, rotate anti-clockwise |

Table 8: Detailed explanation of symbols that we use in question generation with question templates.

