# OpenReview forum: "SoftPhy: Soft-Body Physical Concept  Learning  and Reasoning from Videos"
_ICLR.cc/2024/Conference — Submitted to ICLR 2024_

### Official Review · Reviewer_P1S1 · 2023-10-30

**Soundness:** 2 fair
**Presentation:** 2 fair
**Contribution:** 2 fair
**Rating:** 5
**Confidence:** 5

**Summary:**

The paper introduces the Soft-Body Physical Dataset (SOPHY), a new benchmark for testing AI's physical reasoning with soft objects. SOPHY covers various physical properties like mass and density in dynamic scenarios. Despite the comprehensive nature of the dataset, current AI models show limited performance on it, revealing a gap in their understanding of soft object dynamics. The authors aim for SOPHY to drive improvements in AI's physical world perception and reasoning.

**Strengths:**

1.	This paper targets a really interesting problem, the motivation is sound.
2.	The paper is well-written and easy to follow.

**Weaknesses:**

1.	On the conceptualization of this work. From the bottom of my heart, I like the topic this paper discusses. But since it concerns physics understanding, the description of physics should have a high standard, at least, the very basic concept should be coherent:

a.	In Intro – Second Paragraph. The examples given for humans are very irrelevant to the “soft body”, the topic of this paper. For the liquid example, it can demonstrate the density, but for the pulley example, which physics parameters are the humans trying to distinguish? Moreover, humans can distinguish these physics properties do no means AI models can, so there is a logic leap between the human examples to “However, it remains an …”

b.	From this paragraph on, I notice that the paper has a confusing meaning of “soft body”, how is liquid a type of soft body? According to Wikipedia (Soft-body dynamics - Wikipedia), it should be a solid object at least. Yes, I can find more rigorous sources (e.g. a textbook), but I think a simple checkup on Wikipedia can avoid such concept mistakes.

c.	According to Section 3 Dataset, the paper mentions physical properties: mass, friction, elasticity, density, deformability, and stretchiness.
First, there’s a difference between the physical properties a physics simulator can simulate and a property with genuine physics meaning. Sometimes the physics simulator just combine many underlying physics process and expose some high-level properties for the game developer or artist to control. Here is exactly the case. For example, physically speaking, elasticity and stretchiness are both two types of deformability. I wonder how they can be put at the same level. Besides, stretchiness and deformability are both parameters without corresponding basic physics meaning, which means you cannot measure them in the real world. How would the authors measure the stretchiness of cloth from the real world, as is the method of measurement coherent with what’s inside the physics engine? Besides, for soft body objects, there are more physics properties to influence the deformation such as plasticity, viscosity, etc.

2.	On the writing: The pie charts of Fig 3 and Fig 5-7 do not give precise ratio numbers. The labels of the sections of the pie charts are not clear enough. Take Fig 3 for example, it has “Mass”, “Mass Change”, and “Mass Goal”, then why it does not have “Shape”, “Shape Change”, or “Shape Goal”, if the shape is not a physical property, then why no “Tension”, “Tension Change”, “Tension Goal”. A similar confusion goes for Fig 5-7.

3.	On the dataset: The number of videos is not large. A potential reason is the lack of variance in the scene setup. The soft body can deform in infinite ways, how can a 500-video dataset satisfy the coverage of dynamics?

4.	On the experiments: Will the model train on the proposed dataset be generalized to real-world videos? Or is there any potential way the paper aims for real-world physics reasoning?

**Questions:**

For the questions, please see the Weakness section.

---

> ### Author Response · Authors · 2023-11-18
> **Look forward for your further feedback! (Part 1)**
>
> Thank you for the constructive comments!
>
> **Q1(a). About Pulley**
>
> > [P1S1] a. In Intro – Second Paragraph. The examples given for humans are very irrelevant to the “soft body”, the topic of this paper. For the liquid example, it can demonstrate the density, but for the pulley example, which physics parameters are the humans trying to distinguish?
>
> We are sorry for the potential confusion regarding the liquid and rope (pulley) example. Generally, in this work, we emphasize both the parametric side and the behavioral side of the soft-bodies. In detail, we address this question from two perspectives.
>
> **[Why we choose pulley system as the basis of *rope* scenario?]** We believe, as for human perception, rope is very different from liquid in that rope is 1-dimensional. When we think of rope, we mainly consider the connectivity and length constraints in ropes as well as the derivative dynamics. Viewing liquid dynamics is quite different when we may focus more on the physical properties and its global contour behaviors. In the *rope(aka. pulley)* dataset, we attempt to guide models to learn the embedded physical mechanics and constraints behind a variety of objects. To some extent, lift-up and push-down, cooperating with pulleys, are the most common use of rope in both everyday life(e.g. gym) and industry(e.g. crane). For a simple case, humans could easily perceive the motion of another side of the rope if we see one side of the rope lifted up or pushed down, which is proven diffcult for many VLMs due to dataset gap. As for a more complex case, we know that the dynamic pulley in Fig. 1 (c) of the paper could reduce the tension between ropes compared with static pulley, helping us to pull the cargo up more easily. Hence, we think it's crucial to equip vision language models with a conceptual perception of these phenomena, which is more related to physical dynamics reasoning, as opposed to physical parameter or concept reasoning. For each scenario, we consciously include both reasoning sides.
>
> **[What physical parameters relevant to "soft-body" we are trying to distinguish?]** Typical physical parameters of rope are like elastic coefficient and bending stiffness. Since these parameters are already displayed in the scenario of *cloth* or *ball*, where the parameters are shown in a more comprehensive and complicated way, thus considering the diversity of physical parameters in our overall dataset, in the *rope* scenario, we mainly focus on other physical characteristics of this kind of material. One is the relative weight of load attached to the rope, indicating the external force exerted to the rope terminals, the other is the relative tension force within the rope segments. We hope these two physical parameters can be ideal testing criteria for the conceptual/parametric perception of rope.
>
> **Q1(b). About Logic Leap**
>
> > [P1S1] Moreover, humans can distinguish these physics properties do no means AI models can, so there is a logic leap between the human examples to “However, it remains an …”
>
> Thank you for your concern. In this paragraph, we express concerns about whether current AI models have the capability to accurately identify distinctions in the physical properties and dynamics of soft-body objects, which humans can intuitively discern. To improve our writing, we show the revised paragraph below and have incorporated it in our revision of paper.
>
> *We are able to know that the clear liquid in Fig.1(a) at the bottom has a higher density than the yellow liquid on the top; we know that the dynamic pulley in Fig.1(c) could reduce the tension between ropes compared with static pulley, helping us to pull the cargo up more easily. These innate human skills raise an intriguing question：can current AI models have the physical common sense to infer soft objects' physical properties and predict their corresponding dynamics?*
>
> **Q2. About Definition of Soft-Body**
>
> > [P1S1] b. From this paragraph on, I notice that the paper has a confusing meaning of “soft body”, how is liquid a type of soft body? According to Wikipedia (Soft-body dynamics), it should be a solid object at least. Yes, I can find more rigorous sources (e.g. a textbook), but I think a simple checkup on Wikipedia can avoid such concept mistakes.
>
> The Wikipedia focuses on **Soft-body dynamics, a specialized area within computer graphics**. This term is used to differentiate soft-body dynamics from fluid dynamics, distinct in computer graphics simulation process. While our emphasis is on physical reasoning, not graphic methods. We adopt this concept to underline that prior benchmarks primarily featured rigid bodies, like spheres and cubes. Fluids, being neither rigid nor adequately addressed previously, are a separate and worthwile consideration.
>
> We hope our clarification can resolve your concerns on the "soft body". If there are still concerns on the confusion of the name or any suggestion, we can discuss more and revise the whole paper correspondingly.

---

> > ### Author Response · Authors · 2023-11-18
> > **Look forward for your further feedback! (Part 2)**
> >
> > **Q3. About Physical Parameters**
> >
> > **[Q3(a). Why are different levels of concepts put together? Some are high-level / APIs.]**
> >
> > > [P1S1] c. According to Section 3 Dataset, the paper mentions physical properties: mass, friction, elasticity, density, deformability, and stretchiness. First, there’s a difference between the physical properties a physics simulator can simulate and a property with genuine physics meaning. Sometimes the physics simulator just combine many underlying physics process and expose some high-level properties for the game developer or artist to control. Here is exactly the case. For example, physically speaking, elasticity and stretchiness are both two types of deformability. I wonder how they can be put at the same level.
> >
> > We appreciate your opinions. They are meaningful thoughts from the view of serious physics. However, our thoughts are as follows.
> >
> > 1. **Concept's familiarity to common humans.** The reason why we choose those properties/concepts as the target concepts lies in the motivation of this research. We want to discuss those concepts that are on the intersection of both serious physics and psychology. That is, we explore the perception of common physical concepts on both human beings and machine models, which means human beings should be familiar with these concepts. Otherwise the human study would be undoable. We understand your concern that some concepts discussed in this work are only high-level abstract or integration of one or more basic physical parameters, such as stretchiness and elasticity. But we are afraid common people, especially those without physics background, are able to distinguish some of the most basic physical concepts such as Young's modulus, Cauchy stress, Lamé parameter, and so forth. To be specific, we list some relative concepts about different materials as a table below:
> >
> > | Material  | Common Physical Concept      | Description | Physical or High-Level | (For common human) Familiarity | Is Included |
> > | --------- | ---------------------------- | ----------- | ---------------------- | ------------------------------ | ----------- |
> > | Rope      | Mass                         | Property    | **Physical**           | **Familiar**                   | Y           |
> > | Rope      | Tension                      | Force       | **Physical**           | **Familiar**                   | Y           |
> > | Fluid     | Density                      | Property    | **Physical**           | **Familiar**                   | Y           |
> > | Fluid     | Viscosity                    | Property    | **Physical**           | **Familiar**                   | N |
> > | Cloth     | Streching Stiffness          | Property    | **High-Level**         | **Familiar**                   | Y           |
> > | Cloth     | Bending Stiffness            | Property    | **High-Level**         | **Familiar**                   | Y           |
> > | Cloth     | Friction                     | Force       | **High-Level**         | **Familiar**                   | Y           |
> > | Cloth     | Dynamic Friction Coefficient | Property    | **Physical**           | **Not Sure**                   | **N**       |
> > | Cloth     | Young's Modulus              | Property    | **Physical**           | **Unfamiliar**                 | **N**       |
> > | Soft Ball | Elasticity                   | Property    | **High-Level**         | **Familiar**                   | Y           |
> > | Soft Ball | Plasticity                   | Property    | **High-Level**         | **Familiar**                   | Y           |
> > | Soft Ball | Young's Modulus              | Property    | **Physical**           | **Unfamiliar**                 | **N**       |
> > | Soft Ball | Poisson's Ratio              | Property    | **Physical**           | **Unfamiliar**                 | **N**       |
> > | Soft Ball | Yield Limit                  | Property    | **Physical**           | **Unfamiliar**                 | **N**       |
> >
> > It's shown that we include the human-familiar concepts to fit this cognitive AI research motivation.

---

> > > ### Comment · Reviewer_P1S1 · 2023-11-21
> > > **Reply to part 2**
> > >
> > > Thanks for your response.
> > >
> > > So, I gather the physics in this work is more leaning toward intuitive physics or cognitive physics. I accept narrowing the scope, but I strongly advise making it clearer. Because when this work becomes a popular benchmark, and expose to a wider audience, they might think researchers in the AI field are not rigorous.

---

> > ### Comment · Reviewer_P1S1 · 2023-11-21
> > **Reply to part1**
> >
> > Thanks for your response.
> >
> > I will start with the concept of the soft body. First, "dynamics" is a serious physics term, which describes/studies the physics phenomenon related to Newton's second law. "soft body dynamics" is the dynamics of a soft body. And a soft body is solid. Liquid is not solid. These are just physics facts that have nothing to do with the CG community or else. I am sorry if I put up this Wikipedia page and potentially misled you about the real problem. All I want to say is that liquid is not a kind of soft object. If you want to unite the concept of soft body and liquid, you can refer to continuum mechanics (https://en.wikipedia.org/wiki/Continuum_mechanics). Basically, it distinguishes solid objects (rigid, articulated, deformable/soft) from liquid objects.
> > For a rigorous description, I recommend textbooks in continuum mechanics, for example, introduction to continuum mechanics (https://www.sciencedirect.com/book/9780750685603/introduction-to-continuum-mechanics).
> >
> > So, if this work is for continuum objects, then it should involve rigid and articulated objects. If this work is for the solid object, then it should not have liquid. But the scope of this work slices the concept of solid objects and combines it with liquid, I don't understand how the categorization is done.
> >
> > Even put in the context of computer graphics, not strictly physics, I have found no place to state liquid is the soft body.

---

> ### Author Response · Authors · 2023-11-18
> **Look forward for your further feedback! (Part 3)**
>
> 2. **Former works are in support of this combination.** Concepts such as deformability and elasticity, which is highly abstract and on the different descriptional levels, have been equally proposed and discussed in a series of former cognitive AI research. For example, in Physion++([A]), it includes mass, deformability and elasticity and put them at the same level (see ref[A] Fig.1&2). The parameter levels discussed in our work are proposed similarly as Physion++([A]).
>
> [A] Tung, Hsiao-Yu, et al. "Physion++: Evaluating physical scene understanding that requires online inference of different physical properties." NeurIPS (2023).
>
> 3. **High level properties are easy to both understand and control.** Using APIs from physics engine facilitates the control of diverse parameters and allows us to change different settings in different videos, while it would be inconvenient for us to modify the underlying parameters and to distribute them in a good manner(e.g. cloths are 50% likely to be very rough and 50% likely to be very smooth). Besides, we think that using APIs also assures that the concepts are familiar to humans, which tells why they are extracted as APIs.
>
> **[Q3(b). Question: How to measure high level properties in real world and align with the methods used in simulation?]**
>
> > [P1S1] c. ...Besides, stretchiness and deformability are both parameters without corresponding basic physics meaning, which means you cannot measure them in the real world. How would the authors measure the stretchiness of cloth from the real world, as is the method of measurement coherent with what’s inside the physics engine?
>
>
>
> 1. **These high-level concepts are valuable in cognitive AI research.** As we mentioned above, in this work, we focus on those concepts acceptable for common people, like stretchiness, plasticity, etc. Just as you kindly stated, they are kind of subjective and can hardly be strictly defined in physics and measured in the real world. But as our human study shows, most people can distinguish and apply these physical concepts, which means these concepts are valuable to discuss in cognitive AI field. **We hope models can understand these high-level concepts like humans.**
>
> 2. **It's worth noting that, in this work, we focus more on fuzzy judgment (e.g. which one's X is much greater), not accurate prediction.** Just as you kindly pointed out, these concepts like stretchiness can hardly be strictly defined by any physics formula, so we choose to design fuzzy judgment tasks. And we think measuring the accurate values might not be an urgent problem. In detail, the physical parameter values in the sampling pool are not diversely specified. Typically, for most parameters, we only set 2 or 3 values to sample from and they are greatly different. For example, for a cloth's stretching stiffness, we set 2 values, one value very high and the other very low, to make the cloth appear either difficult to stretch or highly elastic, so that the property difference can be easily captured by humans at the first glance. Thus, in this work, we focus on the agent's ability to distinguish properties, rather than predict numerical values. At least at this visual-language AI research stage, we believe if models fail to do simple fuzzy judgment in the simulation dataset, they may hardly predict the values as well, regardless of whether it is simulated or real-world.
>
> 3. **Although the properties are not measured from simulation (but preset values), we can still find the closest physics parameters and measure them using the same method both in simulation and real world.** Take stretchiness as an example, we think people may agree that it shares a very close meaning with Young's modulus which has strict physical definition. So if we need to collect data in the real world, maybe we can take Young's modulus as a criteria of stretchiness.
>     In mechanics of materials,
>     - Formula of Young's Modulus: $E = \frac{\sigma}{\varepsilon}$
>         - Where $E$ is Young's Modulus, $\sigma$ is stress, and $\varepsilon$ is strain
>
>     - Stress ($\sigma$): $\sigma = \frac{F}{A}$
>         - Where $\sigma$ is stress, $F$ is the force applied, and $A$ is the area of the surface on which the force is acting.
>
>     - Strain ($\varepsilon$): $\varepsilon = \frac{\Delta L}{L_0}$
>         - Where $\varepsilon$ is strain, $\Delta L$ is the change in length, and $L_0$ is the original length.
>
>     Since in the dataset we record the **2D/3D data** (image, segmentation, point cloud, mesh, etc.) for each frame, we can follow the above formulas to measure the property value just like scientists do in the real world.

---

> > ### Author Response · Authors · 2023-11-18
> > **Look forward for your further feedback! (Part 4)**
> >
> > 4. **In the real world, we can still coarsely measure or compare the relative value of these properties.** On the basis of the above explanation, that in this work, we care about the relative property relations between 2 objects, accurate measurement of these parameters is not neccessary. Since the values are highly separated, if needed, we may measure the stretchiness of cloths in the ways of fuzzy measurement, or relative measurement, that we can stretch them under the same external force and measure which one stretches more. In this way, it seems we do not need the accurate value.
> >
> >
> > 5. **Good performance on simulation dataset is the first step to the real world learning and deployment.** We also think it is a little early to think about how to transfer the dataset into real world. From the result of human study on this dataset, the properties shown in the videos are proven to be understandable for humans. This means human eyes are able to distinguish these properties by comparing the behaviors of different objects. However at this stage, VLMs are unable to achieve comparable performance, even though SOPHY is from simulation, not real-world. One step at a time, we hope one day, when VLMs are able to "compare" the properties, then it is time we need to consider the problem of how to teach them to "predict" the properties.
> >
> >
> > 6. **Similar practice has been adopted by former related studies and they used similar simulator to build up dataset.** We would like to show that many valuable studies in recent years also used similar physics engines and algorithms(PBD, MPM...) to build up physical reasoning scenarios, and mainly focused on the cognitive sides of physical reasoning, e.g. paper Physion++ ([A]), Physion ([B]), and ComPhy ([C]). Although they did not consider the real-world transfer (measurement etc.), we agree that generalizing the dataset to the real wild world is very promising and valuable.
> >
> > 7. **Sim-real alignment remains a significant, unresolved challenge in academia.** **Recent studies have indicated potential progress, whereas their simplistic test setups highlight ample opportunities for further improvement.** In aligning simulation methods with reality, algorithms like PBD and MPM in physics engines only approximate physical laws to minimize computational load. This leads to only rough estimations of real-world values. Additionally, even identical properties can yield different visual outcomes in simulations with large time steps. **While recent studies have shown promise in utilizing learning methods to align simulation and real world, achieving state-of-the-art results, e.g. PAC-NeRF ([D]) and NCLaw ([E]), they still fall short, particularly with simpler test setups compared to our more complex dataset.** PAC-NeRF ([D]) shows that there is still much room for the performance improvement. In SOPHY, multiple complex collision events and prolonged time duration may cause significant butterfly effect between simulation and groundtruth. Thus, accurately applying our dataset to real-world scenarios remains a significant challenge.
> >
> >
> > **[Q3\(c\). Question: How can we measure more properties in the real world? (How to consider more properties such as viscosity)]**
> >
> > > [P1S1] c. ...Besides, for soft body objects, there are more physics properties to influence the deformation such as plasticity, viscosity, etc.
> >
> > 1. **Just as we mentioned above**, since in this work we only require humans to do **fuzzy judgment**, more physics parameters can be measured **in a coarse way** to get a relative sense of "which one is greater". Sometimes even eye-measurement could be enough. For example, for honey and oil in real world, we can shake them and easily distinguish which one's viscosity is greater, without instruments. So is plasticity. This is the idea we hold in this work-to make the behavioural distinction obvious by enlarging the gap between different values of specific property.
> >
> > 2. **(In case you might wonder how we handle more properties in the dataset.)** For other properties we do not consider in this work, like the surface tension of fluid, we keep the values diverse to make the phenomena more various, examining the agent's ability to decouple different properties in video input.
> >
> >
> >
> > [A] Tung, Hsiao-Yu, et al. "Physion++: Evaluating physical scene understanding that requires online inference of different physical properties." NeurIPS (2023).
> >
> > [B] Bear, Daniel M., et al. “Physion: Evaluating physical prediction from vision in humans and machines.” NeurIPS (2021).
> >
> > [C] Chen, Zhenfang, et al. “ComPhy: Compositional physical reasoning of objects and events from videos.” NeurIPS (2022).
> >
> > [D] Li, Xuan, et al. “PAC-NeRF: Physics augmented continuum neural radiance fields for geometry-agnostic system identification.” ICLR (2023).
> >
> > [E] Ma, Pingchuan, et al. “Learning Neural Constitutive Laws From Motion Observations for Generalizable PDE Dynamics.” ICML (2023).

---

> > > ### Author Response · Authors · 2023-11-18
> > > **Look forward for your further feedback! (Part 5)**
> > >
> > > **Q4. About Writing**
> > >
> > > > [P1S1] On the writing: The pie charts of Fig 3 and Fig 5-7 do not give precise ratio numbers. The labels of the sections of the pie charts are not clear enough. Take Fig 3 for example, it has “Mass”, “Mass Change”, and “Mass Goal”, then why it does not have “Shape”, “Shape Change”, or “Shape Goal”, if the shape is not a physical property, then why no “Tension”, “Tension Change”, “Tension Goal”. A similar confusion goes for Fig 5-7.
> > >
> > > Thank you for all these suggestions.
> > >
> > > **[Q4(a). Improvement of Pie Charts]** We have updated all pie charts in our paper, incorporating a nested style, precise ratio numbers, and enhanced visual aesthetics. Since rebuttal comments here cannot display images, we kindly ask the reviewer to refer to Figures 3, 5, 6, and 7 in our revised paper.
> > >
> > > **[Q4(b). Names of Questions]** The names assigned to questions are arbitrary and are not related to question designs. We use names as symbols to differentiate each question type. In the pulley scenario, mass is the primary factor that diverses these videos. Rope tension results from object mass, and the objects' shape does not influence the motion and dynamics of the videos. Therefore, these two factors do not contribute to dynamics, making it impractical to design dynamic questions, accounting for why we only design factual questions for them. In general, names are solely for identification purposes. We suggest the reviewer to check Table 3-6 for how we design each question type.
> > >
> > >
> > > **Q5. About Dataset**
> > >
> > > > [P1S1] On the dataset: The number of videos is not large. A potential reason is the lack of variance in the scene setup. The soft body can deform in infinite ways, how can a 500-video dataset satisfy the coverage of dynamics?
> > >
> > > Actually, we do not want to build a training set to encompass all similar situations in the test set. Instead, our goal is to encourage AI models to generalize to unseen situations like human from **limited example demonstrations**. However, we can scale up the data size if necessary due to the dataset's synthetic nature.

---

> > > > ### Author Response · Authors · 2023-11-18
> > > > **Look forward for your further feedback! (Part 6)**
> > > >
> > > > **Q6. About Experiments**
> > > >
> > > > > [P1S1] On the experiments: Will the model train on the proposed dataset be generalized to real-world videos? Or is there any potential way the paper aims for real-world physics reasoning?
> > > >
> > > > **[Transferring Has Been a Hot Discussed Issue]** The ability of transferring to real-world has been a hot discussed issue in related works, like Physion [A] and ComPhy [B]. We agree that models trained on SoPhy are unlikely to generalize to real-world settings **directly**. However, we consider our primary objective to be independent of simulation-to-reality transfer, apparantly a more challenging and board problem. Our benchmark is mainly designed for providing a testing ground for AI models, as we believe that models with great performance in real-world should also excel in SoPhy. Successfully soving problems in simulation videos is the first step on the road to extend it to the real world.
> > > >
> > > > **[Recent Possible Solutions]** Sim-real alignment remains a significant, unresolved challenge in academia. Recent studies have indicated potential progress, whereas their simplistic test setups highlight ample opportunities for further improvement. In aligning simulation methods with reality, algorithms like PBD and MPM in physics engines only approximate physical laws to minimize computational load. This leads to only rough estimations of real-world values. Additionally, even identical properties can yield different visual outcomes in simulations with large time steps. While recent studies have shown promise in utilizing learning methods to align simulation and real world, achieving state-of-the-art results, e.g. [C] and [D], they still fall short, particularly with simpler test setups compared to our more complex dataset. [C] shows that there is still much room for the performance improvement. In SOPHY, multiple complex collision events and prolonged time duration may cause significant butterfly effect between simulation and groundtruth. Thus, accurately applying our dataset to real-world scenarios remains a significant challenge.
> > > >
> > > >
> > > > **[More Sensor Data to Support]** Given the huge advancement and limitations in soft-body physical reasoning of Vision&Languege models, our benchmark is particularly important for advancing the area. To bridge the gap between simulation and real-world, we plan to release additional sensor data beyond standard 2D visual perceptions. This aims to provide a more holistic view of the environmental settings, and encourage researchers to challenge with this issue. Diverse sensor data is crucial to step towards more powerful AI models with practical understanding of soft-body physical properties and interactions.
> > > >
> > > > [A] Bear, Daniel M., et al. "Physion: Evaluating physical prediction from vision in humans and machines." NeurIPS (2023).
> > > >
> > > > [B] Chen, Zhenfang, et al. "ComPhy: Compositional physical reasoning of objects and events from videos." NeurIPS (2022).
> > > >
> > > > [C] Li, Xuan, et al. “PAC-NeRF: Physics augmented continuum neural radiance fields for geometry-agnostic system identification.” ICLR (2023).
> > > >
> > > > [D] Ma, Pingchuan, et al. “Learning Neural Constitutive Laws From Motion Observations for Generalizable PDE Dynamics.” ICML (2023).

---

> > > > > ### Author Response · Authors · 2023-11-21
> > > > > **Look forward to more discussions**
> > > > >
> > > > > Dear Reviewer #P1S1
> > > > >
> > > > > We appreciate your constructive suggestions and comments. With the discussion deadline approaching, we are prepared to provide any additional clarifications required. In our revised manuscript, we have thoroughly examined your comments, incorporating extensive analysis and discussion to address your suggestions. We hope that these additional explanations can underscore the merits of our work. Should you require further clarifications or additional experiments, please feel free to reach out.
> > > > >
> > > > > Regards,
> > > > >
> > > > > Authors

---

> ### Comment · Reviewer_P1S1 · 2023-11-21
> **Adjust rating after taking into account the authors and other reviewers' feedback**
>
> During rebuttal, the authors have made clear the scope of the "physics" in this work. Given this change of precondition, I can relax the questions on the rigor of physics. But it still does not justify including liquid objects in soft objects, and I see no possibility to fix this.
> Thus, I raise my rating to 5.

---

> ### Author Response · Authors · 2023-11-22
> **Thanks for your constructive comments and suggestions!**
>
> Thank you for your insightful suggestions! We agree with your idea that physical concepts should be clearer and more rigorous. Therefore, we've broadened our work's description **from 'soft body' to 'continuum'** to more accurately reflect its scope.
>
> In both physics and computer graphics, 'continuum' encompasses various bodies such as liquids, soft materials (e.g., soft balls, cloth, and ropes), rigid bodies (e.g., cubes, pillars, plates, and spheres), and articulated bodies (e.g., pulleys). Our dataset comprehensively encompasses fluids, soft bodies, rigid bodies, and articulated bodies (e.g., in the rope scenario). This inclusion leads to our utilization of the continuum concept, enhancing the breadth and relevance of our study.
>
> Moreover, we consciously include both physical dynamics reasoning (e.g., interactions between fluids, soft bodies and rigid bodies), and physical parameter or concept reasoning (e.g., density for fluids; tension, elasticity for soft bodies; mass for rigid bodies). For instance, our rope and pulley scenario involves elements of rope, rigid bodies, and articulated bodies; the fluid scenario includes liquids; the cloth scenario covers both cloth and rigid bodies; and the ball scenario focuses on soft balls. This extensive coverage ensures that our dataset provides a comprehensive understanding of the interactions and couplings within these various types of continua, capturing the complexity and diversity of real-world physical phenomena.
>
> In light of this, we have renamed the dataset **from SoPhy to ContPhy**. We have added an entire section in **Appendix A.4** to elaborate on the concept of the continuum. We have also **updated the entire paper accordingly**.
>
> We hope this clarification addresses all your concerns. We are looking forward to your further reply and suggestions. **If you find our response satisfactory, we would be grateful if you could consider revising your score.**

---

### Official Review · Reviewer_Vp2g · 2023-10-31

**Soundness:** 2 fair
**Presentation:** 2 fair
**Contribution:** 3 good
**Rating:** 3
**Confidence:** 4

**Summary:**

The authors present the Soft-Body Physical Dataset (SOPHY), an innovative benchmark designed to assess machine learning models' capacity for physical reasoning within a range of scenarios involving soft bodies. The authors subsequently assessed several visual models, such as CNN and MAC, using the dataset. Their findings suggest that contemporary AI models are yet to fully grasp the physical commonsense associated with soft objects, underscoring the significance of the introduced dataset.

**Strengths:**

The paper boasts several commendable attributes. Foremost, the dataset it introduces is characterized by a notable diversity in its scenarios, offering a comprehensive spectrum for analysis. Furthermore, the evaluation of the properties associated with soft objects is designed with meticulous detail. Another significant strength is the decent render quality, which not only enhances the visual clarity but also aids in the accurate interpretation of data. Moreover, the paper provides a comprehensive comparison between human perception, random/frequent answer, non-visual model, and other visual models.

**Weaknesses:**

- Some results are just stated without further discussion.
  - In Section 4.2 Paragraph "Physical Property", The author stated that "ALPRO achieves the best results in the rope scenario, and maintains competitive results in other scenarios, showing the value of large-scale video-text pre-training and alignment.". However, why other models slightly outperform ALPRO in scenarios other than Rope is not discussed.
  - In Section 4.2 Paragraph "Dynamics", only the result of ALPRO and HCRN is discussed, and why other models do not work well is missing.
  - In Section 4.2 Paragraph "Scenario Analysis.", only cloth and rope scenarios are discussed.

- The writing could benefit from some improvements.
  - In Section 4.2 Paragraph 1, "We summarize the performance of all baselines in Table 1.", should be Table 2.

- Some statement is not supported well.
  - In Section 4.2 Paragraph "Evaluation Conclusion", the authors concluded that "Machine models results show that even state-of-the-art models struggle with answering physical questions based on the visual input.". However, the relationship between "soft body physics reasoning capability" and "answering physical questions based on visual input" is not clear. For example, AI models may understand soft body physics well, but unable to understand the questions, as the semantic information is not recognized by the model.
  - In conclusion, the authors stated that "Despite progress, our evaluation of AI models revealed an ongoing challenge: they struggle to perform well on our benchmark, highlighting their limited physical commonsense for soft objects.", but there are already articles(i.e. [1]) concluded that AI models lack physical reasoning capability, if this capability is missing, the model should also lack physical reasoning capability for soft objects.

[1] Li, Shiqian & Wu, Kewen & Zhang, Chi & Zhu, Yixin. (2022). On the Learning Mechanisms in Physical Reasoning.

**Questions:**

- Should tasks related to reasoning on liquids and soft objects be evaluated in different ways? The comprehension of physics for soft objects and liquids may represent divergent capabilities.

---

> ### Author Response · Authors · 2023-11-18
> **Look forward for your further feedback! (Part 1)**
>
> Thank you for all these constructive comments!
>
> **Q1. About Further Discussion of Experiments**
>
> Thank you for all these suggestions! We discuss each question below, and have added them in Section 4.2 of our revised paper.
>
> > [Vp2g] In Section 4.2 Paragraph "Physical Property", The author stated that "ALPRO achieves the best results in the rope scenario, and maintains competitive results in other scenarios, showing the value of large-scale video-text pre-training and alignment.". However, why other models slightly outperform ALPRO in scenarios other than Rope is not discussed.
>
> **[Q1 (a). Other Models Slightly Outperform ALPRO]** Actually, all the baseline models struggle to achieve decent performance on physical property questions, except ALPRO, which achieves the best results in the rope scenario and maintains competitive results in other scenarios, showing the advantages of large-scale video-text pre-training and alignment.
>
> > [Vp2g] In Section 4.2 Paragraph "Dynamics", only the result of ALPRO and HCRN is discussed, and why other models do not work well is missing.
>
> **[Q1 (b). Other Baselines]** Traditional pure neural networks have difficulty understanding the physical scenarios and in capturing the physical laws from videos and question-answer pairs. Thus, they perform worse than their previous performance on our benchmark.
>
> > [Vp2g] In Section 4.2 Paragraph "Scenario Analysis.", only cloth and rope scenarios are discussed.
>
> **[Q1 \(c\). Other Scenarios]** ALPRO performs well in the rope and cloth scenario. Except for the reason that the cloth and rope scenarios share some similarities, another important reason is the fewer question types in the rope and cloth scenario than those in the fluid and ball scenarios. Specifically, the rope scenario has counterfactual and goal-driven, and the cloth scenario has predictive. Conversely, in the fluid and ball scenarios, we incorporated all four problem types, thereby making situations much more complicated. To effectively address these scenarios, models must tackle four distinct question types, each focusing different aspects of physical dynamics. Consequently, no baseline models can gain an absolute advantage in these scenarios. This indicates that our four proposed question types well evaluate different dimensions of physical reasoning, making the fluid and ball scenarios particularly challenging for AI models.
>
> All these comments have been updated in Section 4.2 of our revised paper to further analyze the experiments.
>
>
> **Q2. Reference of Table**
>
> > [Vp2g] In Section 4.2 Paragraph 1, "We summarize the performance of all baselines in Table 1.", should be Table 2.
>
> Thanks for the reminder. We have already fixed this and checked the whole paper again to improve our writing.
>
> **Q3. About Concerns for Semantic Information**
>
> > [Vp2g] In Section 4.2 Paragraph "Evaluation Conclusion", the authors concluded that "Machine models results show that even state-of-the-art models struggle with answering physical questions based on the visual input.". However, the relationship between "soft body physics reasoning capability" and "answering physical questions based on visual input" is not clear. For example, AI models may understand soft body physics well, but unable to understand the questions, as the semantic information is not recognized by the model.
>
> Admittedly, semantic information can influence results. However, using QA formats to evaluate physical reasoning abilities has been widely acknowledged and shown effectiveness in previous works, such as ComPhy [A] and CLEVRER [B]. QA gives the flexibility to estimate different kinds of physical reasoning abilities such as understanding different physical properties like mass, density, viscosity, and predict the corresponding dynamics based on variance of such physical properties. To correctly answer these questions, AI models need to have both the physical common sense and the ability to understand natural language queries and answers. Since the question-answer pairs in our benchmark are synthesized by templates, the major challenge of AI models is from physical reasoning. Thus, we claim that our SoPhy benchmark mainly evaluates "soft-body physical reasoning capability".
>
> [A] Chen, Zhenfang, et al. "ComPhy: Compositional physical reasoning of objects and events from videos." NeurIPS (2022).
>
> [B] Yi, Kexin, et al. "CLEVRER: Collision events for video representation and reasoning." ICLR (2020).

---

> > ### Author Response · Authors · 2023-11-18
> > **Look forward for your further feedback! (Part 2)**
> >
> > **Q4. About AI Models Lack Physical Reasoning Capability**
> >
> > > [Vp2g] In conclusion, the authors stated that "Despite progress, our evaluation of AI models revealed an ongoing challenge: they struggle to perform well on our benchmark, highlighting their limited physical commonsense for soft objects.", but there are already articles(i.e. [1]) concluded that AI models lack physical reasoning capability, if this capability is missing, the model should also lack physical reasoning capability for soft objects.
> >
> > Thank you very much for your suggestions. We appreciate the conclusion from the paper [C] that dynamics-based models may not accurately predict and AI models lack physical reasoning capability. However, on the other hand, several methods [D, E, F, G] have successfully tackled previous benchmarks like Physion [A] and CLEVRER [B]. In paper [D], VQA accuracy for CLEVRER is 96.5%, and for Physion is 67.1%, which is very close to human performance. These papers have successfully validated the performance of state-of-the-art AI models. Motivated by their significant contribution, we aim to extend this success further by evaluating dynamics and soft-body objects. **Additionally, we have discussed this paper [C] in the related work section of the revised paper.**
> >
> > [A] Bear, Daniel M., et al. "Physion: Evaluating physical prediction from vision in humans and machines." NeurIPS (2023).
> >
> > [B] Yi, Kexin, et al. "CLEVRER: Collision events for video representation and reasoning." ICLR (2020).
> >
> > [C] Li, Shiqian, et al. "On the Learning Mechanisms in Physical Reasoning." NeurIPS (2022).
> >
> > [D] Wu, Ziyi, et al. "Slotformer: Unsupervised visual dynamics simulation with object-centric models." ICLR (2023).
> >
> > [E] Ding, David, et al. "Attention over learned object embeddings enables complex visual reasoning." NeurIPS (2021).
> >
> > [F] Ding, Mingyu, et al. "Dynamic visual reasoning by learning differentiable physics models from video and language." NeurIPS (2021).
> >
> > [G] Lu, Haoyu, et al. "VDT: General-purpose Video Diffusion Transformers via Mask Modeling." Arxiv (2023).

---

> > > ### Author Response · Authors · 2023-11-21
> > > **Look forward to your feedback!**
> > >
> > > Dear Reviewer #Vp2g
> > >
> > > Thanks again for your constructive suggestions and comments. As the deadline for discussion is approaching, we are glad to provide any additional clarifications that you may need.
> > >
> > > In our previous response, we have carefully studied your comments and added a lot more analysis and discussions to complement your suggestions.
> > >
> > > We hope that the provided additional explanations have convinced you of the merits of our work. Please do not hesitate to contact us if there are other clarifications or experiments we can offer.

---

> > > > ### Author Response · Authors · 2023-11-22
> > > > **Look forward to your further feedback!**
> > > >
> > > > Dear Reviewer #Vp2g
> > > >
> > > > We appreciate your constructive suggestions and comments.
> > > >
> > > > With the discussion deadline approaching in one day, we are prepared to offer any further clarifications required.
> > > > Should further clarifications or additional experiments be necessary, please feel free to request them. If no concerns, would you kindly consider raising the score?
> > > > Thanks again for your very constructive and insightful feedback!
> > > >
> > > > Regards,
> > > > Authors

---

> > > > > ### Author Response · Authors · 2023-11-22
> > > > >
> > > > > Dear Reviewer,
> > > > >
> > > > > We would like to thank you for your helpful feedback which has helped us improve the paper.
> > > > >
> > > > > We addressed reviewers' concerns in the author's responses several days ago. We would be delighted if you could please take a look at our detailed responses so that we can address any remaining concerns before the end of the discussion phase.
> > > > >
> > > > > Sincerely,
> > > > >
> > > > > Authors

---

### Official Review · Reviewer_U8tp · 2023-11-01

**Soundness:** 3 good
**Presentation:** 3 good
**Contribution:** 3 good
**Rating:** 6
**Confidence:** 2

**Summary:**

The paper introduces a new dataset, / and benchmark for targeting to assess machine learning models in physical reasoning.
The paper explains how this dataset is complementary o existing datasets.

**Strengths:**

The motivation behind creating SPHY is to advance ML / AI techniques to bridge the gap between human and AI in the physical world. The authors generated results for several benchmarks.

**Weaknesses:**

NA

**Questions:**

NA

---

> ### Author Response · Authors · 2023-11-18
> **Look forward for your further feedback! (Part 1)**
>
> Thank you for the constructive comments!
>
> We are glad to hear that you think our SOPHY dataset is novel and invaluable, which advances ML / AI techniques to bridge the gap between human and AI in the physical world.
>
> We are looking forward for your further feedback!

---

### Official Review · Reviewer_t1H7 · 2023-11-01

**Soundness:** 3 good
**Presentation:** 4 excellent
**Contribution:** 3 good
**Rating:** 6
**Confidence:** 4

**Summary:**

This paper presents SOPHY a new soft-body benchmark including four types of simulated videos (based on Unity) and their corresponding question-answering pairs which can serve as a new benchmark to study AI models on understanding complex physical properties and dynamics for soft-body scenarios. The paper also evaluate the performance of several SotA methods and show that there is still a lot room to improve as they fall behind human performance.

**Strengths:**

- the paper proposes a new benchmark that involves careful task environment designs and question-answering pairs generation, which is technically novel and interesting.
- the proposed four types of soft-body tasks are indeed lacking from existing benchmarks and they are more complex so the proposed benchmark adds values to the community.
- the authors benchmarked several SotA methods on the proposed benchmark, provided good analysis, conducted human performance study, and showed that there is still a lot room to do research, which are all quite valuable to the community.

**Weaknesses:**

- the task family is limited to the designed four types. Also the questions are generated from pre-defined sets of templates. These restrict the general use of the benchmark for other tasks, environments, and questions. Could the authors comment on how is it possible to extend the framework for other tasks?
- the authors claimed that previous benchmarks cannot change mass and friction, but as many of them are also based on physical simulators, it's unclear why they couldn't do that.
- the paper doesn't propose a solution to improve the performance based on the findings.

**Questions:**

see weakness

---

> ### Author Response · Authors · 2023-11-18
> **Look forward for your further feedback! (Part 1)**
>
> Thank you for all these constructive comments!
>
> **Q1. About Extending the Framework for Other Tasks**
>
> > [t1H7] the task family is limited to the designed four types. Also the questions are generated from pre-defined sets of templates. These restrict the general use of the benchmark for other tasks, environments, and questions. Could the authors comment on how is it possible to extend the framework for other tasks?
>
> We have developed various templates for each question type across four distinct scenarios. Our experiments reveal that current AI models fail in understanding soft-body physical dynamics, leading to poor performance on our dataset, independent of template design. Additionally, our dataset generation framework is capable of producing different question types and video types, such video annotations for segmentation and depth. This framework can support a range of common vision tasks, such as video segmentation, object detection, text-video retrieval and video grounding. Since our primary objective is to evaluate physical reasoning abilities in the form of question answering, we will not prioritize other vision tasks.
>
> **Q2. About Previous Benchmarks**
>
> > [t1H7] the authors claimed that previous benchmarks cannot change mass and friction, but as many of them are also based on physical simulators, it's unclear why they couldn't do that.
>
> We agree with the reviewer that many prior benchmarks, such as Physion [A] and ComPhy [B], are also based on physical simulators. However, their benchmarks fall short in thoroughly assessing whether machine models have human-like visual reasoning abilities, particularly in understanding physical object properties and dynamics. These physical benchmarks do not adequately cover dynamics influenced by varying physical object properties across diverse scenarios, notably in the case of soft objects. Some of them lack the variance of object parameters, such as Physion [A], in which solids share the same mass and water maintains a consistent density. Some of them limit their scope to simple primitives and scenario, such as ComPhy [B], in which they only consider mass and electric charge in a basic solid collision scenario of cubes and spheres. In contrast, we are the only comprehensive benchmark with object-specific properties in a wide range of scenarios, which even encompasses soft-body objects and unique language-based questions about dynamics in counterfactual and goal-planning scenarios. Our dataset addresses the gap in existing benchmarks, which often **overlook complex intrinsic properties** and **corresponding dynamics** in diverse scenarios, a key aspect our SoPhy emphasizes.
>
> [A] Bear, Daniel M., et al. "Physion: Evaluating physical prediction from vision in humans and machines." NeurIPS (2023).
>
> [B] Chen, Zhenfang, et al. "ComPhy: Compositional physical reasoning of objects and events from videos." NeurIPS (2022).
>
>
> **Q3. About Proposed Solution**
>
> > [t1H7] the paper doesn't propose a solution to improve the performance based on the findings.
>
> We thank the reviewer for this suggestion. We have presented a proposed solution for the fluid scenairo to better investigate the characterstics of the current benchmark in **G2** of the **General Response**. Please kindly refer to it and we are looking forward to your further feedbacks!

---

> > ### Author Response · Authors · 2023-11-22
> > **Looking forward to your further discussion.**
> >
> > Dear Reviewer t1H7,
> >
> > We sincerely thank you for your valuable feedback on our manuscript. Following your suggestions, we have enriched our paper with additional experiments and discussion with related work, which are now included in the revised manuscript.
> >
> > As the rebuttal period concludes, we hope our efforts align with your expectations. If you find our response satisfactory, we would be grateful if you could consider revising your score.
> >
> > Thank you once again for your insightful guidance.
> >
> > Warm regards,
> >
> > Authors

---

### Author Response · Authors · 2023-11-18
**General Response: Contributions, New Proposed Oracle Model and Paper Revision (Part 1)**

**[G1. Contribution Recognition]**

We thank reviewers for their thoughtful feedback. We are glad that reviewers recognized the following contributions.

- **Task**. The task is novel and well-motivated.

> "technically novel and interesting" (t1H7)

> "The motivation behind creating SOPHY is to advance ML / AI techniques to bridge the gap between human and AI in the physical world" (U8tp)

> "an innovative benchmark" (Vp2g)

> "targets a really interesting problem, the motivation is sound" (P1S1)

- **Benchmark**. The benchmark is invaluable and high-quality.

> "the proposed four types of soft-body tasks are indeed lacking from existing benchmarks" (t1H7)

> "The authors generated results for several benchmarks" (U8tp)

> "Another significant strength is the decent render quality, which not only enhances the visual clarity but also aids in the accurate interpretation of data" (Vp2g)

> "the evaluation of the properties associated with soft objects is designed with meticulous detail" (Vp2g)

- **Experiments**. Experiments are comprehensive and valuable.

> "the authors benchmarked several SotA methods on the proposed benchmark, provided good analysis, conducted human performance study, and showed that there is still a lot room to do research" (t1H7)

> "The authors generated results for several benchmarks" (U8tp)

> "the paper provides a comprehensive comparison between human perception, random/frequent answer, non-visual model, and other visual models" (Vp2g)


**[G2. New Oracle Model: SoPRO]**

In addition to the point-wise responses below, we add a new solution method supporting experiments in the rebuttal according to reviewers’ suggestions.

We introduce **So**ft-Body **P**hysical **R**easoning **O**racle Model, **SoPRO**, which marries physical-based dynamics models with the recent large language models which enjoy the advantages of both models, precise dynamic predictions, and interpretable reasoning. We discuss the overall neural-symbolic framework in **G2(a)**, the large language model backbone in **G2(b)**, the visual perception model in **G2\(c\)**, and the physical simulation model in **G2(d)**. We also present the representative results and analysis in **G2(e)**.

**We kindly recommend reviewers and ACs to refer to A.3 SoPRO: Soft-Body Physical Reasoning Oracle Model in Appendix of our revised paper for more details!**

**[G2(a). Neural-Symbolic Framework]**
Inspired by prior works [A, B, C], we propose an oracle neural-symbolic framework named SoPRO. As the rapid development of large language models and their applications, we replace previous rule-based neural executor with the newest neural-symbolic large language models. Furthermore, our oracle model is equipped with a state-of-the-art differentiable physical simulator，and the popular visual perception model, MASK R-CNN [F]. This physical simulator uses point cloud as initial input, which is hard to be perceived directly from videos. That accounts for the reason that we call it **oracle** model.

[A] Chen, Zhenfang, et al. "ComPhy: Compositional physical reasoning of objects and events from videos." NeurIPS (2022).

[B] Yi, Kexin, et al. "CLEVRER: Collision events for video representation and reasoning." ICLR (2020).

[C] Yi, Kexin, et al. "Neural-symbolic VQA: Disentangling reasoning from vision and language understanding." NeurIPS (2018).

[F] He, Kaiming, et al. "Mask R-CNN." ICCV (2017).

**[G2(b). Large Language Model]**
Similar to ViperGPT [E], we utilize large language models, specifically ChatGPT [M], as the backbone of our proposed SoPRO. We further develop a set of dynamics modules and visual perception modules serving as APIs for code generation model. With the provided API access and a pre-defined physical reasoning prompt, we leverage ChatGPT to generate Python code that can be directly executed and interpreted. This method bridge the gap between language comprehension and physical concept understanding, enabling us to directly evaluate the soft-body physical reasoning abilities of AI models, without concerns related to different expressions of question sentences.

[E] Surís, Dídac, et al. "ViperGPT: Visual inference via python execution for reasoning." ICCV (2023).

[M] Ouyang, Long, et al. "Training language models to follow instructions with human feedback." NeurIPS (2022).

**[G2\(c\). Visual Perception Model]**
We adopt MASK R-CNN [F] as our video perception module. This module aids SoPRO to detect and ground objects in videos, bridging the gap between linguistic descriptions of objects and their physical properties. Given an input video, this module will detect objects in each frame, addressing simple property questions directly, such as object counting, object existence and color recognition, with very high accuracy. Also its ability to locate objects through bounding boxes indirectly plays a role in other questions.

[F] He, Kaiming, et al. "Mask R-CNN." ICCV (2017).

---

> ### Author Response · Authors · 2023-11-18
> **General Response: Contributions, New Proposed Oracle Model and Paper Revision (Part 2)**
>
> **[G2(d). Physical Simulation Model]**
> Previous methods [A, K, L] mainly adopt DPI-Net as physical simulator, which is a 3D particle-based graph neural network [G]. DPI-Net performs well in mass [L], while falls short in fluid dynamics prediction. Therefore, we utilize a differentiable MPM physics simulation module DiffMPM [N] to provide a primary solution to the system identification of the *fluid* scenario. We also borrowed the simulation framework partly from PAC-NeRF [H].
>
> [A] Chen, Zhenfang, et al. "ComPhy: Compositional physical reasoning of objects and events from videos." NeurIPS (2022).
>
> [G] Li, Yunzhu, et al. "Learning particle dynamics for manipulating rigid bodies, deformable objects, and fluids." ICLR (2019).
>
> [H] Li, Xuan, et al. "PAC-NeRF: Physics augmented continuum neural radiance fields for geometry-agnostic system identification." ICLR (2023).
>
> [K] Bear, Daniel M., et al. "Physion: Evaluating physical prediction from vision in humans and machines." NeurIPS (2023).
>
> [L] Tung, Hsiao-Yu, et al. "Physion++: Evaluating physical scene understanding that requires online inference of different physical properties." NeurIPS (2023).
>
> [N] Hu, Yuanming, et al. "DiffTaichi: Differentiable programming for physical simulation." ICLR (2020).
>
>
> **[G2(e). Result Analysis]**
> |            | Property | Counter. opt. | Counter. ques. | Goal. opt. | Goal. ques. | Pred. opt. | Pred. ques. |
> | :--------- | :------: | :-----------: | :------------: | :--------: | :---------: | :--------: | :---------: |
> | Random     |   33.3   |     52.9      |      6.0       |    59.9    |     7.5     |    53.8    |     4.8     |
> | Frequent   |   52.7   |     57.9      |      17.2      |    63.1    |    36.3     |    50.1    |    12.5     |
> | Blind-LSTM |   49.3   |     56.1      |      7.8       |    57.3    |    22.5     |    51.4    |    12.5     |
> | CNN-LSTM   |   54.0   |     55.0      |      8.6       |    57.3    |    22.5     |    51.4    |    12.5     |
> | MAC        |   30.0   |     56.5      |      6.9       |    51.2    |    17.5     |    53.5    |    12.5     |
> | HCRN       |   52.7   |     52.6      |      4.3       |    67.7    |    41.3     |    50.6    |     1.9     |
> | ALPRO      |   48.0   |     56.8      |      6.0       |    62.7    |    32.5     |    53.8    |    12.7     |
> | **SoPRO-Oracle** | **78.0** |   **75.7**    |    **36.2**    |  **77.3**  |  **60.0**   |  **90.1**  |  **68.3**   |
>
> The physical simulation method [H] in oracle model exhibits powerful performance that can precisely simulate soft-body dynamics, while it requires scenario-specific manipulation of hyperparameters. Therefore, we present representative results in the fluid scenarios below, which encompasses all four question types, and compare them with multiple baseline models. We report the representative results in the table above, which reveals that our SoPRO successfully outperforms all questions in the fluid scenario.
>
> In property questions, SoPRO impresses with a 78.0% overall accuracy. This category includes two tasks: Stick Number which is much more simpler and Density which is more complex. The high accuracy in Stick Number significantly raises SoPRO's overall performance in property questions, making it a standout model. In dynamics questions, SoPRO surpasses all baselines, particularly in predictive questions, hitting a 68.3% accuracy rate. This is because simulator's objective focuses on predicting dynamics during its training.
>
> [H] Li, Xuan, et al. "PAC-NeRF: Physics augmented continuum neural radiance fields for geometry-agnostic system identification." ICLR (2023).
>
>
> **[G2(f). Conclusion]**
> We have proposed a new oracle model SoPRO, that marries physical-based dynamics models with large language models to reason on soft-body physical concept. We evaluate SoPRO in the fluid scenario and analyze comprehensively. The great performance of SoPRO-Oracle, with the initial input of cloud points that cannot be directly perceived, serves as a guide and oracle for future AI models.
>
> **Note that we only summarize each section in this comment. We have already incorporated the detailed method as a new section in Appendix A.3.**
>
>
> **[G3. Revision Summary]**
>
> We made the following modifications in our revision to address reviewers' questions (highlighted in blue in the revision):
>
> - Oracle model as our solution to this benchmark (t1H7)
> - New nested pie charts with ratio numbers for better visualization (P1S1)
> - New citation and discussion of paper [J] (Vp2g)
> - Correct reference of Table 1 (Vp2g) and improved writing (P1S1)
>
> [J] Li, Shiqian, et al. "On the Learning Mechanisms in Physical Reasoning." NeurIPS (2022).

---

### Author Response · Authors · 2023-11-20
**General response (update): thanks for the comments and look forward to post-rebuttal feedbacks!**

Thanks again for all of your constructive suggestions, which have helped us improve the quality and clarity of the paper!

Since there are less than three days to the end of the discussion period, we have not heard any post-rebuttal response yet.

Please don’t hesitate to let us know if we can offer any additional clarifications or experiments, as we would love to convince you of the merits of the paper.

We appreciate your suggestions.

---

### Meta-Review · Area_Chair_rnZi · 2023-12-14

**Metareview:**

While the proposed soft body physics reasoning benchmark offers some intriguing aspects, reviewers raise critical concerns about its scientific validity, scope, and limitations. These issues, particularly the confusion surrounding the inclusion of liquids and the lack of rigorous physics representation, led to my reject recommendation.

The benchmark introduces a diverse range of soft body tasks not found in existing benchmarks, potentially pushing the boundaries of AI physics understanding. The paper provides a comprehensive evaluation framework, including analysis of physical properties, dynamics, and scenario variations. Also, comparing AI models to human performance adds valuable insights into the limitations of current models.

However, as mentioned by reviewers, and I agree, the paper uses simplified physical properties and parameters without clear connections to real-world physics. Terms like "stretchiness" and "deformability" lack precise definitions and measurable counterparts in the real world. The 500-video dataset unlikely offers sufficient diversity and complexity to capture the full spectrum of soft body dynamics. The paper doesn't address how models trained on this benchmark would perform on real-world physics reasoning tasks.

**Justification For Why Not Higher Score:**

While the authors present an interesting idea, the rigorousness of the presentation needs to be improved significantly. Also, as suggested, please revise the scope to focus solely on solid soft bodies, addressing the confusion surrounding liquids. Moreover, there is a need to expand the dataset size and introduce more variation to improve its representativeness.

Lastly, please improve the clarity and detail of the writing and presentation, particularly in results analysis and figure design.

**Justification For Why Not Lower Score:**

N/A

---

### Decision · Program_Chairs · 2024-01-16

Reject